# Spatial regulation of Drosophila ovarian Follicle Stem Cell division rates and cell cycle transitions

David Melamed[1], Aaron Choi[1], Amy Reilein[1], Simon Tavaré[1,2], Daniel Kalderon[1] *

**1** Department of Biological Sciences, Columbia University, New York, New York State, United States of America, **2** Irving Institute for Cancer Dynamics & Department of Statistics, Columbia University, New York, New York State, United States of America

* ddk1@columbia.edu

**Data Availability Statement:** All relevant data are within the manuscript and its Supporting Information files.

**Funding:** This work was supported by NIH RO1 GM079351 to DK. The funders had no role in study

## Abstract

Drosophila ovarian Follicle Stem Cells (FSCs) present a favorable paradigm for understanding how stem cell division and differentiation are balanced in communities where those activities are independent. FSCs also allow exploration of how this balance is integrated with spatial stem cell heterogeneity. Posterior FSCs become proliferative Follicle Cells (FCs), while anterior FSCs become quiescent Escort Cells (ECs) at about one fourth the frequency. A single stem cell can nevertheless produce both FCs and ECs because it can move between anterior and posterior locations. Studies based on EdU incorporation to approximate division rates suggested that posterior FSCs divide faster than anterior FSCs. However, direct measures of cell cycle times are required to ascertain whether FC output requires a net flow of FSCs from anterior to posterior. Here, by using live imaging and FUCCI cell-cycle reporters, we measured absolute division rates. We found that posterior FSCs cycle more than three times faster than anterior FSCs and produced sufficient new cells to match FC production. H2B-RFP dilution studies supported different cycling rates according to A/P location and facilitated live imaging, showing A/P exchange of FSCs in both directions, consistent with the dynamic equilibrium inferred from division rate measurements. Inversely graded Wnt and JAK-STAT pathway signals regulate FSC differentiation to ECs and FCs. JAK-STAT promotes both differentiation to FCs and FSC cycling, affording some coordination of these activities. When JAK-STAT signaling was manipulated to be spatially uniform, the ratio of posterior to anterior division rates was reduced but remained substantial, showing that graded JAK-STAT signaling only partly explains the graded cycling of FSCs. By using FUCCI markers, we found a prominent G2/M cycling restriction of posterior FSCs together with an A/P graded G1/S restriction, that JAK-STAT signaling promotes both G1/S and G2/M transitions, and that PI3 kinase signaling principally stimulates the G2/M transition.

design, data collection and analysis, decision to publish, or preparation of the manuscript.

**Competing interests:** No competing interests

## Author summary

Adult stem cells reside in various tissues in the body and replenish specialized cells throughout an organism's lifetime. Tissues control the rate of stem cell division to match differentiated cell loss by sending signals to the stem cells. We used Drosophila ovarian follicle stem cells as a model to measure stem cell division rates and how they are regulated. We accomplished this by imaging the stem cells in living and fixed tissue using genetically introduced fluorescent markers of progress through different phases of the cell cycle. The length of these phases (DNA replication, mitosis and the intervening preparation stages known as G1 and G2) were measured for stem cells in different locations and with genetic changes simulating changes in natural signals. The results showed large differences in division rates according to the precise location of stem cells, which are partly explained by a specific graded signal and its action at two phase transitions in the cell cycle. The measured division rates fit a model where the stem cells in two different locations produce different specialized cells at different rates but also exchange locations, with roughly equal flow in each direction, to produce a single dynamic stem cell community.

## Introduction

Adult stem cells must regulate their rates of division and differentiation in order to supply appropriate numbers of derivative cells without depleting the stem cell reservoir [1–3]. Stem cell division and differentiation are not temporally coupled for several important highly proliferative stem cell populations, including mammalian gut and epidermal skin stem cells, as well as Drosophila FSCs; instead, each process can be regulated independently [4–6]. In these systems of "population asymmetry" [7–9], individual stem cells and their progeny continually compete for survival and amplification, according to apparently stochastic decisions of whether to divide or differentiate. Genetic changes in an individual stem cell that only increase its division rate necessarily produce a competitive advantage for its lineage; such mutations may therefore commonly be a first step towards cancer, amplifying a pre-cancerous stem cell population [4,10–14]. Clearly, understanding how division rate is regulated in stem cell paradigms where differentiation is independent of division is extremely important for uncovering how a stem cell pool is maintained over a lifetime and how specific genetic changes can be oncogenic.

Drosophila FSCs provide an outstanding paradigm to explore the regulation of stem cell proliferation for several reasons. First, it has been demonstrated that altered division rates have a profound effect on stem cell survival and amplification, in agreement with the theoretical expectations outlined above [4]. Second, the location and behavior of FSCs have now been clearly defined, correcting earlier mis-conceptions [2,10], and third, several extracellular signals that regulate FSC division have been identified. Specifically, Hedgehog (Hh), JAK-STAT (Janus Kinase- Signal Transducer and Activator of Transcription) and PI3K (Phospho-Inositide 3' Kinase) pathways were shown to stimulate FSC division and make FSCs more competitive, as were several mediators of cell cycle transitions, including CycE (CyclinE), Stg (String-ortholog of yeast Cdc25 [Cell division Control 25]) and Yki (Yorkie) [10,15–21].

In contrast to stem cell paradigms where transitions from long-term quiescence into cycling have been the focus of investigations of proliferation [22–24], a key issue for FSCs and similar constitutively active stem cells in mammalian gut or epidermis is how the rate of division of continuously cycling stem cells is regulated. The opportunity to explore regulation of stem cell cycling is further enriched in the FSC paradigm by pronounced spatial modulation; posterior

FSCs appear to cycle faster than anterior FSCs and three major signaling pathways (Hh, Wnt and JAK-STAT) are known to be graded over the FSC domain [10,18,25–27].

It is likely that the distribution of extracellular signals can change in response to specific stresses to allow stem cells to adapt to specific situations; indeed, the spread of Hh in the Drosophila germarium is known to depend on nutritional status [15,16]. Here we are exploring standard, well-fed laboratory conditions to understand how FSCs are maintained while replacing differentiated cells with high constitutive turnover.

FSCs exhibit an additional intriguing feature of producing different derivatives at different locations. The location of FSCs was determined by examining FC-producing lineages with only a single candidate stem cell position [10]. The total number of somatic cells in the locations identified is similar to the number of FSCs per germarium deduced by lineage analyses. Those analyses counted the number of different multicolor lineages in a single ovariole or the proportion of all FCs in an ovariole contributed by a single FSC lineage, measured over a variety of time periods to account for the loss of FSC lineages over time [4,10]. A ring of about eight "layer 1" FSCs, midway along the germarium and immediately anterior to a key spatial landmark of strong Fasciclin 3 (Fas3) surface protein staining, directly give rise to about 5–6 proliferative FCs during each 12h cycle of egg chamber production. The number of FCs produced per cycle has been deduced from both the number of different multi-colored FSC lineages in a single egg chamber and the average fraction of all FCs in a single egg chamber derived from a single contributing FSC lineage [4,10]. Anterior and immediately adjacent to layer 1 FSCs are layer 2 FSCs (six, on average) (Fig 1A). These anterior FSCs and their layer 3 FSC neighbors (two, on average) directly give rise to 1–2 quiescent ECs every 12h on average [4]. Anterior and posterior FSCs also exchange positions over time, as deduced from the distribution of pairs of descendants 3d after marking single FSCs, so that a single lineage can contribute both FCs and ECs [10]. This organization, termed "dynamic heterogeneity", supports a stable FSC population, maintained within a narrow A/P (anterior-posterior) domain, while continuously supplying both FCs posteriorly and ECs anteriorly. The signals that govern FSC behavior must therefore coordinate FSC division rate with two types of differentiation, occurring in different locations and at different rates.

FSCs lie at the intersection of opposing gradients of Wnt and JAK-STAT pathway signaling (Fig 1A) [10,26,27]. The magnitude of each pathway governs the propensity of a FSC to move anterior or posterior within the FSC domain or to become an EC or an FC. Higher JAK-STAT signaling favors posterior movement and differentiation to an FC, while higher Wnt signaling favors anterior movement and differentiation to an EC [10,18]. Wnt signaling does not appear to influence FSC division rate, in contrast to JAK-STAT [18].

Since posterior FSCs are depleted by becoming FCs at a much higher rate (roughly four-fold) than anterior FSCs become ECs, there would be a pronounced net flow of FSCs from anterior to posterior positions if all FSCs divided at the same rate. However, posterior FSCs appear to divide faster than anterior FSCs by a factor of about 1.7, based on the proportion of cells in S-phase, measured by incorporation of the thymidine analog EdU (ethynyl deoxyuridine) [10,18]. Because EdU incorporation does not suffice to measure division rates, it remains to be assessed whether the difference in division rates quantitatively matches the different rates of depletion of anterior and posterior FSCs. If the rates match, there would be little or no net A/P (anterior-posterior) flux and FSCs in all locations would have equivalent potential to share the collective stem cell burden. In sum, FSC proliferative control is particularly interesting because both the A/P pattern and absolute rates of FSC division are important.

JAK-STAT pathway activity is stimulated by a ligand (Upd; Unpaired) released from newly-formed polar FCs and is graded across the FSC domain, with lowest activity in ECs (Fig 1A) [26]. The magnitude of JAK-STAT pathway activity has a large effect on division rate,

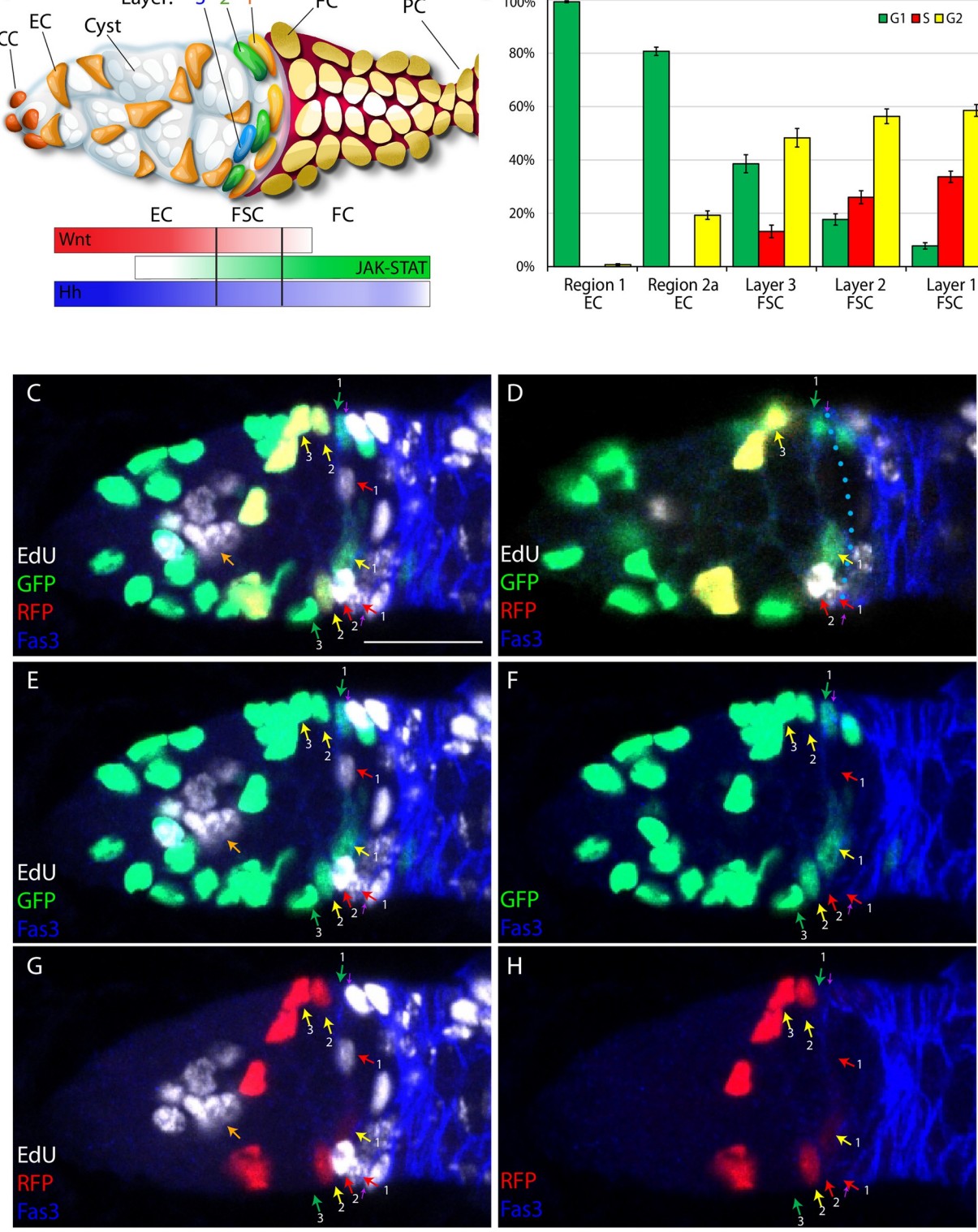

**Fig 1. FUCCI reporter and EdU labeling to score cell cycle phase of all FSCs and ECs.** (A) Cartoon representation of a germarium based on [18]. Cap cells (CCs) at the anterior (left) contact germline stem cells (not highlighted), which produce cystoblast daughters that mature into 16-cell germline cysts (white) as they progress posteriorly. Quiescent Escort cells (ECs) extend processes around germline cysts and support their differentiation. Follicle Stem Cells (FSCs) occupy three A/P rings (3, 2, 1) around the germarial circumference and immediately anterior to strong Fas3 expression (red) on the surface of all early Follicle Cells (FCs). FCs proliferate to form a monolayer epithelium, including specialized terminal

Polar cells (PCs), which secrete the Upd ligand responsible for generating a JAK-STAT pathway gradient (green). Wnt pathway (red) and Hh pathway (blue) gradients have opposite polarity and are generated by ligands produced in CCs and ECs. (B) Percentage of cells in the designated locations in G1 (green), S (red) and G2/M (yellow), deduced from FUCCI reporters in wild-type germaria. SEM is shown from scoring 28 germaria (574 r1 ECS, 638 r2a ECs, 205 layer 3 FSCs, 323 layer 2 FSCs, 490 layer 1 FSCs). (C-H) The same *C587>FUCCI* germarium after EdU (white) labeling is shown using (C, E-H) several superimposed z-sections to capture roughly half of all ECs and FSCs present or (D) a single z-section to illustrate how cell locations are scored (see also Materials and Methods and S1 Fig). Cells expressing GFP-only (green arrows), RFP-only (red arrows) or both (yellow arrows) in (C) are clarified by images of the same set of z-sections showing only (E, F) GFP or only (G, H) RFP channels, with (E, F) or without (F, H) the EdU channel (Fas3 is present in all). All three FSCs in S-phase (red arrows; white EdU label) express neither GFP nor RFP. Many FCs, posterior to FSCs (further right), and a germline cyst (large clustered nuclei; orange arrow) have (white) EdU label. The anterior border of strong Fas3 staining is indicated by small purple arrows at the edge of the germarium. Sample are scored by examining each z-section, as shown in (D). Here, the Fas3 border has been outlined (blue dotted line), immediately adjacent to the nuclei of a G1 (green) layer 1 FSC at the top, a G2 (GFP plus RFP- clear only if GFP channel omitted) layer 1 FSC towards the bottom, and an S-phase layer 1 FSC (faint white from EdU because most of the nucleus is in an adjacent section) at the bottom. An S-phase layer 2 cell lies one more cell anterior (bottom), while a G2 (yellow) layer 3 FSC is two cells from layer 1 (top). Scale bar is 10μm. Raw data are in S1 Data (tab Fig 1B).

indicated by EdU incorporation frequencies and FSC accumulation, and appears to be a major contributor to the A/P pattern of proliferation [18]. Further understanding of how FSC division is balanced with differentiation and how division rate is regulated might benefit from better measures of cell cycling to supplement EdU studies, and by ascertaining which cell cycle transitions are regulated by JAK-STAT signaling and other inputs. Here we employ the cell cycle reporter, FLY-FUCCI [28] and an H2B-RFP dilution strategy to those ends.

## Results

### FUCCI reporters provide an accurate visual display of FSC cell cycle stages

The distribution of cells among phases of the cell cycle can report which transitions appear to limit cycling. FLY-FUCCI reporters link an E2F1 degron to EGFP, conferring rapid degradation in S phase, and a CycB (CyclinB) degron to mRFP1, conferring rapid degradation at the end of M phase and during G1 [28]. Thus, "GFP-only" indicates cells in G1, "GFP plus RFP" indicates G2 or M phase (recognized by morphology) cells, while S-phase cells ideally express only RFP. In practice, the onset of detectable RFP after the G1/S transition depends on the speed with which new protein can be made, so early S-phase cells may have no detectable GFP or RFP. Likewise, the addition of GFP to RFP at the start of G2 may have a lag, so that cells in early G2 may only have detectable RFP.

After trials with FUCCI reporters driven directly by a *ubiquitin* gene promoter or driven indirectly (*UAS-FUCCI*) by yeast GAL4 expressed from *act-GAL4*, *tj-GAL4* or *C587-GAL4* transgenes we found that *C587-GAL4* together with two copies of *UAS-FUCCI* reporters produced the strongest signals in FSCs. The *C587-GAL4* expression domain has previously been defined by driving UAS-reporter proteins that are relatively stable (GFP, RFP, β-galactosidase) and reported as being expressed strongly in ECs and declining posteriorly over the FSC domain, with some signal in the earliest FCs [10,29]. FUCCI reporters revealed a similar pattern of robust signals over the entire FSC and EC domains with a major decline in labeling at the anterior border of strong Fas3 staining, beyond which are FCs (Figs 1 and S1). Although some Fas3-positive FCs are labeled, the strong reduction of expression between FSCs and FCs is extremely useful for reporting the border between FSCs and FCs during live imaging, where Fas3 cannot be used as a precise positional marker.

Using *C587-GAL4* and *UAS-FUCCI* reporters together with a 1h pulse of EdU labeling immediately prior to fixation, we found that no cells with GFP-only or GFP-plus-RFP expression included EdU, consistent with assignment of all such cells to G1 and G2 (or M), respectively (Fig 1C–1H). Some EdU-positive cells expressed only RFP but the majority were colorless, suggesting that there is a significant delay in accumulating detectable RFP after the

start of S-phase. All RFP-only cells had EdU label, indicating that GFP accumulates quickly at the end of S-phase. Thus, G1 (GFP only) and G2/M (GFP + RFP) cells are clearly defined, while S-phase cells can be recognized as lacking both GFP and RFP (the majority) or expressing only RFP. Live imaging confirmed the expected sequence of transitions between the above color-coded phases (see later).

We also examined germaria stained with DAPI, instead of EdU incorporation, to visualize each somatic cell nucleus (S1 Fig). The deduced proportion of S-phase cells, counted as colorless or RFP-only (33% in layer 1; 24% in layer 2), was very similar to the proportion of S-phase cells detected directly in numerous EdU-labeled samples from earlier lineage studies (33% in layer 1; 20% in layer 2) [18]. This correspondence is consistent with all cells lacking GFP and RFP being in S-phase. We can therefore deduce that all cells were labeled with either GFP or EdU in experiments omitting DAPI labeling, while the cell cycle stage of all cells can be determined by a combination of GFP, RFP and DAPI in the absence of EdU labeling. We therefore mostly conducted tests with DAPI staining rather than EdU labeling and combined data from the two approaches.

### Major cycling barriers are at G2/M for posterior FSCs and at G1/S for more anterior cells

Consistent with earlier studies using only EdU labeling, the most posterior (layer 1) FSCs were more frequently in S-phase than their anterior neighbors (34% vs 26% for layer 2 and 13% for layer 3), while ECs were never in S-phase (Fig 1B) [10,18]. Layer 1 FSCs were mostly in G2 (59%), with very few in G1 (8%). Layer 2 FSCs had twice the frequency of G1 cells (18%) but the majority were still in G2 (56%), while the proportion of G1 (39%) and G2 cells (48%) were much more similar in layer 3. Region 2a ECs were mostly in G1 (81%) and the most anterior ECs (region 1) were almost entirely in G1 (99%). Clearly, ECs are restricted by an inability to transition into S-phase (and might largely be considered to be in a G0 phase), while the most posterior FSCs pass quickly through G1, with cycling apparently limited principally by a G2/M barrier. There must therefore be a robust A/P gradient of factors that stimulate the G1/S transition or an inverse distribution of G1/S inhibitors.

To test the effects of altered genotypes (Figs 2 and 3) we used GAL4-responsive transgenes, together with *C587-GAL4* and a transgene encoding temperature-sensitive GAL80 protein expression. Flies were raised at the permissive temperature (18C) and shifted to the restrictive temperature (29C) 3d before analysis. Previous tests of the same signal transduction component genes to be investigated here, with the same protocol, found that reporters for signaling pathway activity showed significant changes by 2d and robust changes by 3d at the restrictive temperature [18]. We selected the shortest effective time (3d) in order to study primary responses, prior to any systematic compensatory mechanisms potentially triggered by accumulation or depletion of FSCs. We first explored known cell cycle mediators, before turning to the effects of different signaling pathways.

CycE-Cdk2 (Cyclin-dependent kinase 2) activity is a key determinant of passage into S-phase and can be regulated through the levels of CycE protein or Cdk inhibitors, such as the Dacapo (Dap) protein [30–34]. In Drosophila there is a single *cycE* gene and CycE/Cdk complex, with an essential role in the cell cycle. Provision of additional CycE increased the proportion of FSCs in S-phase for all three FSC layers (Fig 3A–3C), as observed previously when additional CycE, expressed from *tubulin* and *actin* gene promoters, was limited to clones of cells derived from a single FSC [18]. The proportion of cells in G1 was diminished in layer 1 FSCs (from 7.8% to 3.1%), as might be expected for promoting the G1/S transition (Fig 3). The proportional reductions in G1 cells were lower for layer 2 (17.7% to 12.4%) and layer 3 (38.5%

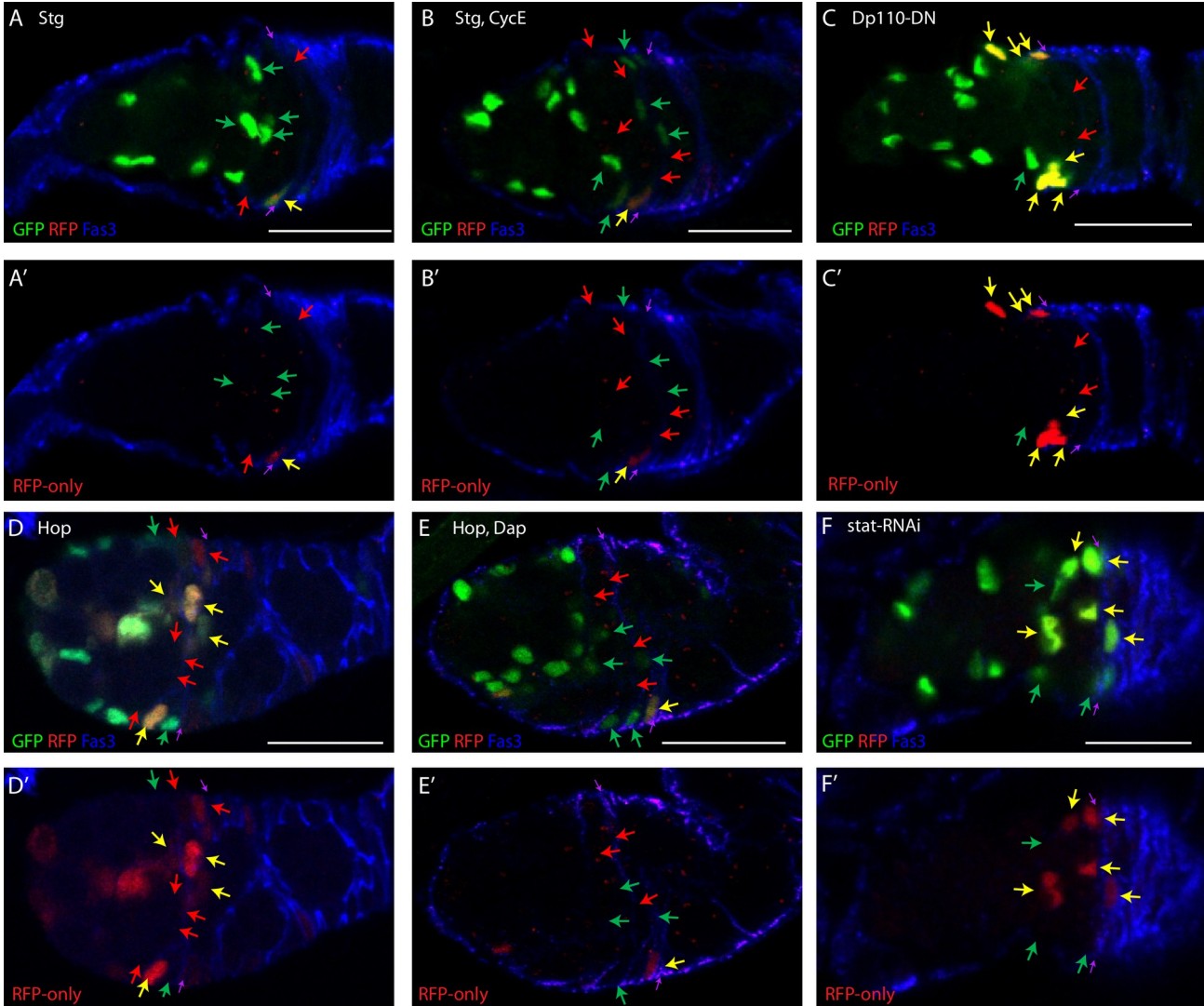

**Fig 2. FUCCI reporter responses to key cell cycle regulators and signals.** Examples of responses of FUCCI reporters to conditional expression of indicated transgenes, showing Fas3 (blue) staining to infer FSC identity and location, (A-H) GFP and RFP or (A'-H") only RFP. All samples were also stained with DAPI to be able to count all cells, including those in S-phase (red arrows). All G1 (green arrows) and G2/M (yellow arrows) FSCs in the range of z-sections shown are indicated, as is the anterior Fas3 border (purple arrows). (A) Excess Stg increased G1 and decreased G2 FSC numbers. (B) Excess Stg together with CycE greatly increased S and decreased G2 FSC numbers. (C) Dominant-negative PI3K decreased G1 and increased G2 FSC numbers. (D) Excess JAK-STAT activity decreased G2 and increased S FSC numbers, but (E) principally increased G1 and decreased G2 FSC numbers together with excess Dacapo, Cdk2 inhibitor. (F) Decreased JAK-STAT activity increased G2 and decreased S FSC numbers. ECs were mainly in G1 in all samples but (D) and (E) illustrate conversion of some to G2 by excess JAK-STAT activity. Scale bar is 10μm.

to 33.3%), with a notably high residue of G1 layer 3 FSCs. There were also small reductions of the proportions of cells in G2 for all FSC layers. The proportion of G1 cells in region 2a (78%) and region 1 ECs (98%) remained substantially unchanged. Thus, although excess CycE appears to promote G1/S transitions in the most posterior FSCs, it is surprisingly ineffective in more anterior FSCs and especially in ECs. This suggests that the G1/S transition in ECs and anterior FSCs is restricted by a factor other than CycE-Cdk2 activity or that the deficit in CycE/Cdk2 activity is too large to be reversed by the strategy we used.

The activity of Stg (String; yeast Cdc25 ortholog) is critical for G2/M passage and is often regulated transcriptionally [35–39]. Additional Stg expression dramatically reduced the

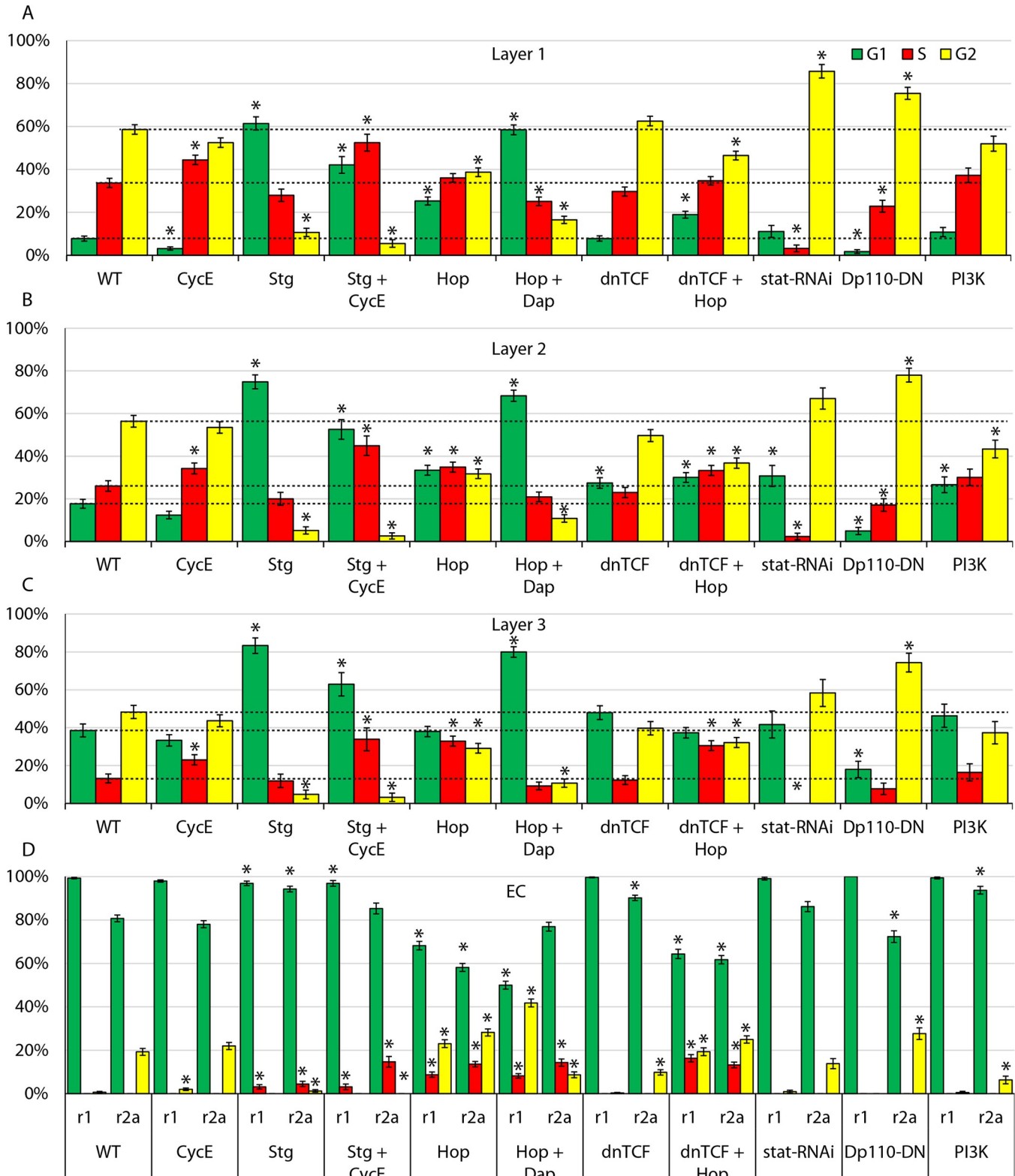

**Fig 3. Quantitative impact of cell cycle regulators and signaling pathways on cell cycle distributions.** (A-D) Percentage of (A) layer 1 FSCs, (B) layer 2 FSCs, (C) layer 3 FSCS, (D) region 2a ECs and region 1 ECs in G1 (green), S (red) and G2/M (yellow), assessed by scoring FUCCI reporters for germaria expressing the indicated transgenes under *C587-GAL4* control in the presence of temperature-sensitive GAL80, after shifting to the restrictive temperature of 29C for 6d (*stat RNAi*) or 3d (all others). SEM is shown. The number of cells scored ranged from 126 to 588 (average of 372) for layer 1 FSCs, 88 to 403 (average

of 255) for layer 2 FSCs, 48 to 308 (average 165) for layer 3 FSCs, 190 to 755 (average 447) for r2a ECS, and 179 to 754 (average 426) for r1a ECs; precise numbers in S1 Data. Horizontal dotted lines indicate control values for G1, S and G2/M; significant differences for each are indicated by an asterisk (p<0.05, n-1 chi-squared test). Raw data are in S1 Data (tabs Figs 3A–3CRawData, 3A–3C and 3A-C p value).

proportion of FSCs in G2 in all layers (59% to 11%, 56% to 5%, 48% to 5%) (Figs 2A and 3), consistent with earlier studies that showed a large increase in the proportion of cells positive for the M-phase marker, phosphor-histone H3 [19]. There was no increase in the proportion of FSCs in S phase, with the cells accelerated through G2 accumulating in G1 (61% in layer 1, 75% in layer 2, 83% in layer 3). Stg overexpression was found previously to shorten G2 and extend G1 in Drosophila wing disc cells; further study suggested that the extension of G1 involved the action of E2F1, which mediated compensatory feedback mechanisms between G1 and G2 when either was artificially altered [40]. When both additional Stg and CycE were expressed in ECs and FSCs, the fraction of FSCs in G2 was also very low (<6%) in all layers but the proportion of cells in S-phase was now substantially higher than controls in all layers (52% vs 34%, 45% vs 26%, 34% vs 13%) (Figs 2B and 3). Thus, in Drosophila FSCs, excess Stg potently promoted mitosis and caused the majority of FSCs to accumulate in G1. Also adding excess CycE significantly reduced this G1 accumulation but the proportion of FSCs in G1 was still much higher than normal for all FSC layers (42% vs 8%, 53% vs 18%, 63% vs 39%). A distinct gradient of increasing S-phase proportions from anterior to posterior remained, again suggesting a spatially graded G1/S restriction that is not readily overcome by excess CycE.

Excess Stg virtually eliminated the small fraction (19%) of region 2a ECs normally in G2 (Fig 3D). Surprisingly, a few region 2a (4%) and region 1 ECs (3%) were now found in S-phase rather than G1. These S-phase fractions were increased to 15% and 12%, respectively, when excess CycE was co-expressed. Thus, entry of a significant proportion of ECs into S-phase requires both excess Stg and CycE. In both ECs and FSCs, excess CycE alone causes significant G2 accumulation and excess Stg causes significant G1 accumulation.

## JAK-STAT signaling promotes G1/S and G2/M transitions

Excess Hop (Hopscotch: Drosophila JAK) substantially decreased the fraction of FSCs in G2 for all layers (59% to 39%, 56% to 32%, 48% to 29%) (Figs 2D and 3A–3C). The fraction of FSCs in S-phase was significantly increased, especially in layer 2 (26% to 35%) and layer 3 (13% to 33%), where *C587-GAL4* is expressed most strongly. The increase in S-phase occupancy differs from the effects of excess Stg, which also depletes G2 cells, and indicates that excess JAK-STAT signaling promotes the G1/S transition as well as strongly accelerating G2 passage.

In region 2a and region 1 ECs, excess JAK-STAT activity reduced the fraction of cells in G1 (81% to 58%, 99% to 68%), while increasing the fraction in S (from 0 to 14% and 9%, respectively) and G2 (19% to 28%, 1% to 23%). Thus, in ECs the most prominent action of increased JAK-STAT is to stimulate G1/S passage.

When the CycE/Cdk2 inhibitor Dacapo (Dap) was co-expressed with *UAS-Hop* the fraction of FSCs in S-phase was reduced in all layers compared to wild-type (34% to 25%, 26% to 21%, 13% to 9%), with the majority of FSCs accumulating in G1 (58%, 68%, 80%) (Figs 2E and 3A–3C). Thus, Dap inhibition of G1/S passage trumped the stimulation observed for excess JAK-STAT pathway activity alone. Surprisingly, however, Dap did not reduce the fraction of ECs driven into S-phase by *UAS-Hop*, even though the distribution of ECs between G1 and G2 was somewhat altered. Thus, promotion of G1/S transitions by increased JAK-STAT pathway activity was phenocopied by excess CycE and blocked by a CycE/Cdk2 inhibitor in FSCs, but not in ECs, suggesting significant mechanistic differences between the regulation of the G1/S

transition in FSCs and ECs. The strong effect of increased JAK-STAT pathway activity on exit from G2 in all FSC layers was even more obvious in the presence of excess Dap (Fig 3A–3C).

Depletion of gene products using GAL4-responsive *UAS-RNAi* transgenes can take longer than increasing protein activities by overexpression. Expression of *stat RNAi* had only small effects on FUCCI reporters after 3d (S1 Data for Fig 3A–3C) but robust effects by 6d, so we conducted FUCCI analysis at 6d. In layer 1 FSCs, the proportion of cells in S-phase was dramatically reduced (34% to 3%), with many additional cells accumulating in G2 (59% to 86%), indicating a major deficit in entering mitosis (Figs 2F and 3A–3C). The fraction of cells in S-phase was also greatly reduced in layer 2 (26% to 2%) and layer 3 (13% to 0) but these anterior FSCs were more equally distributed between G1 (18% to 31%, 39% to 42%) and G2 (56% to 67%, 48% to 58%), suggesting that the loss of JAK-STAT reduced the frequency of both G1/S and G2/M transitions. These observations are consistent with deductions from responses to increased JAK-STAT activity, indicating JAK-STAT promotion of G2/M transitions, most prominently in posterior FSCs, and also G1/S transitions, more prominently in anterior FSCs. The FUCCI profile of ECs, where JAK-STAT activity is normally very low, was barely altered by loss of STAT (Fig 3D).

## Cell cycle transitions are largely unaffected by Wnt signaling

Previous investigation of Wnt pathway reduction or elimination in FSC lineages showed important roles in guiding FSC locations and differentiation, but suggested that normal Wnt signaling does not greatly influence either the magnitude or pattern of cell division, inferred from EdU incorporation [18]. Artificially increasing Wnt pathway activity in FSC lineages, to levels estimated at twice the physiological maximum in the germarium, severely reduced EdU incorporation but this was overridden by excess JAK-STAT pathway activity [18]. Also, Wnt pathway reduction could promote EdU incorporation, in concert with excess CycE, when JAK-STAT activity was eliminated [18]. Thus, Wnt pathway activity appears to have the potential to reduce FSC cycling but that potential was blocked by JAK-STAT pathway activity when assayed simply by EdU incorporation in FSC lineages.

Here, we used expression of dominant-negative TCF (dnTCF; "T-Cell Factor", sole transcriptional effector of Drosophila Wnt signaling) to reduce Wnt signaling over the entire EC/FSC domain. Under these conditions, Wnt signaling revealed by a *Fz3-RFP* reporter is lower than normally found in layer 1 FSCs, and appears to be spatially uniform, contrasting with the normal strong decline from anterior to posterior [18]. Only small changes in FUCCI reporters were observed in FSCs, with slightly more layer 2 FSCs in G1 as the only notable change (Fig 3A–3C). The distribution of FSC cell cycle phases in the presence of excess JAK-STAT pathway activity was also virtually unchanged by additional inhibition of Wnt signaling (*dnTCF + UAS-Hop*; Fig 3A–3C). These results confirm prior conclusions that Wnt signaling under normal conditions, and in the presence of excess JAK-STAT activity, has very little effect on FSC cycling.

In ECs, Wnt inhibition modestly reduced the frequency of region 2a G2 cells (from 19% to 10%); all remaining cells were in G1 (Fig 3D). In the presence of excess JAK-STAT, Wnt inhibition provoked no significant changes in EC cell cycle profiles. Thus, even though Wnt signaling is highest in ECs and has the potential to block FSC cycling [18], the prominent G1/S barrier in ECs does not depend on high Wnt pathway activity.

## PI3K pathway activity promotes entry into mitosis throughout the FSC domain

In Drosophila, insulin-like growth factor binding to the insulin receptor is transmitted via the insulin-like receptor substrates, Chico and Lnk, to activate PI3K [41–44]. As in mammals,

downstream responses to PI3K include Tor complex-mediated translational and metabolic changes and FoxO-dependent transcriptional changes, generally integrated to promote cell growth and proliferation. Unlike in mammals, various other tyrosine kinase receptors, which principally activate Ras/MAPK signaling, do not generally induce significant concerted PI3K pathway activation. When PI3K pathway activity was inhibited by expression of a dominant-negative version of the catalytic subunit, Dp110 (aka PI3K92E, Phospho-Inositide 3' Kinase 92E, here "PI3K"), the fraction of FSCs in G2 was significantly increased in all layers (Figs 2C and 3A–3C). The increase in layer 1 (from 59% to 75%) was less than due to STAT loss (to 86%) but the increases in layers 2 and 3 were greater than observed for STAT loss (56% to 78% vs 67%, and 48% to 74% vs 58%). Thus, PI3K pathway activity appears normally to promote exit from G2 in FSCs and shares this responsibility with JAK-STAT signaling, taking on a greater role in more anterior cells where JAK-STAT pathway activity is normally lower.

PI3K activity reduction caused a bigger fractional decrease in G1 than S-phase occupancy for all FSC layers relative to controls (G1: 8% to 2%, 34% to 23%, and 18% to 5% vs S: 26% to 17%, 39% to 18%, and 13% to 8%), suggesting that PI3K does not normally play a prominent role in G1 exit. This contrasts with a much greater reduction in S-phase than G1 representation when JAK-STAT activity was lost, consistent with JAK-STAT signaling also promoting FSC G1/S transitions.

Increased expression of the PI3K catalytic subunit (using "*UAS-PI3K*") produced converse but smaller responses (only changes in layer 2 FSCs were statistically significant), reducing the fraction of FSCs in G2 in all layers (59% to 52%, 56% to 43%, 48% to 37%), while increasing the fraction of cells in G1 (8% to 11%, 18% to 27%, 39% to 46%) and S-phase (34% to 37%, 26% to 30%, 13% to 16%) to similar degrees (Fig 3). These results show that excess PI3K can increase G2 exit and perhaps also G1 exit (because S-phase frequency was marginally increased, whereas excess Stg reduced the fraction of cells in S-phase). The responses to excess PI3K were much smaller than observed for excess JAK-STAT, especially for anterior FSCs, where only excess JAK-STAT promoted an increase in S-phase cells. Thus, the PI3K pathway appears to stimulate the G2/M transition throughout the FSC domain and may have some, lesser potential to facilitate G1 exit.

G2/M stimulation extended to the EC domain, where increased PI3K activity reduced region 2a cell G2 proportions (from 19% to 6%), but still with no cells in S-phase. Conversely, PI3K pathway inhibition increased the fraction of region 2a G2 ECs to 29%. All region 1 ECs remained in G1 in both genetic conditions. Thus, PI3K pathway activity stimulates G2/M transitions in ECs and FSCs, but has little or no effect on the G1/S transition in ECs or FSCs.

## Live FUCCI imaging of partial cell cycles to infer the duration of cell cycle phases

The proportion of cells of a certain type, or in a restricted location, that label with the mitotic marker, phospho-histone H3, or incorporate EdU as an indicator of S-phase, is often used as a proxy to estimate either the proportion of cycling cells or the average rate of cell cycling. The fraction of cells in M-phase or S-phase under different conditions will, however, only provide a measure of relative cell cycle times if the length of that phase remains constant under the conditions being compared. While regulation of G1/S and G2/M transitions is common, there is no guarantee that the length of M or S phase is constant for a given cell type under different conditions or in different locations. We therefore used FUCCI reporters, together with a previously developed system for live imaging of germaria, freshly dissected from flies and embedded in Matrigel [10,45], to measure absolute cell cycle times.

We deliberately imaged for only short periods of time (generally close to 2h; average 148min) to capture normal *in vivo* behavior, even though viable, active germaria can be

imaged for as long as 8-12h [10,45]. The pattern of *C587-GAL4* expression, mostly terminating posteriorly with layer 1 FSCs (Figs 1 and S1), allowed us to identify layer 1, layer 2 and layer 3 FSCs, even though there was no Fas3 staining landmark. Imaging each z-section every 20 mins on average allowed us to track each labeled cell with confidence. Some cells moved out of the z-section range during imaging, while most cells were tracked for the entire imaging period. We observed each type of expected transition: loss of GFP, indicating a G1/S transition, and entry of G2 cells into mitosis, which was generally observed morphologically in only a single frame, followed by loss of RFP to leave two GFP-only nuclei of daughters in G1 (Fig 4A, S1 Movie). Occasionally, an S to G2 phase transition was observed with GFP signal added to RFP, but the fate of cells initially in S-phase was not systematically tracked because the majority of S-phase cells have no detectable RFP or GFP. Thus, quantitation focused on cells in G1 (GFP only), G2 (GFP plus RFP) and mitosis. Generally, 3–8 FSCs were observed to transition from one phase to another in a single germarium. We therefore excluded data where no transitions from G1 or G2 were observed; often, such germaria exhibited only minimal cell movements, in contrast to apparently healthy germaria.

We reasoned that each cell in a specific phase (say, G2) was captured at an arbitrary time within that phase at the start of imaging. Hence, we can infer the average phase duration for a set of observed cells (such as layer 1 FSCs) by adding together the total time spent by all observed cells in that phase and dividing by the total number of transitions observed out of that phase. The intuitive and mathematical basis for this method of phase duration estimation is discussed extensively in S1 Methods, along with considerations of sufficient and even sampling, as well as estimated errors and the impact of potentially heterogeneous cell cycling behavior (see S1 Methods: "Live FUCCI Calculation Method" and "Sampling Considerations for Live FUCCI Calculation Method"). Collectively, we followed 92 layer 1 FSCs among fifteen germaria, spending a total of 8432 mins in G2 with 16 transitions out of G2 (to give an estimate of G2 as 527 mins), and a total of 1621 mins in G1 with 20 transitions to S phase (to give an estimate of G1 as 81 mins) (Fig 4 and S1 Data). Similarly, the average time spent in M-phase was 17 mins (11 transitions through mitosis). From the observations of fixed wild-type germaria presented earlier, the time spent in S-phase as a fraction of the rest of the cell cycle (G1 +G2+M) was 33.7%/66.3%. Hence, the deduced duration of S phase was (527+81+17) x 33.7/ 66.3 = 318 mins and the average total cell cycle time was therefore 943 mins. The duration of G1 (8.6%) and G2/M (57.7%) as a fraction of the whole cell cycle of layer 1 FSCs from live imaging matched fixed image FUCCI data very well (7.8% G1, 58.6% G2/M), providing some assurance of correct cell location identification and the absence of a major change induced by the live imaging procedure. The total number of cells tracked in different layers (92 in layer 1 = 50%, 72 in layer 2 = 39%, 20 in layer 3 = 11%) also matched the proportion of FSCs known to be in layers 1 (50%), 2 (37%) and 3 (13%), further confirming appropriate cell location identification.

An analogous strategy was used to estimate average layer 2 FSC cell cycle phase times from 72 tracked cells for G1 (407 mins), S (838 mins), G2 (1958 mins), and M (20 mins), summing to 3223 mins (Fig 4C). Only four transitions from G2, and three from G1 were observed in total, so cell cycle time estimates have a much larger margin of error than for layer 1 FSCs. Nevertheless, it is clear that layer 1 FSCs cycle much faster than layer 2 FSCs, estimated as a factor of more than 3-fold (3223/943 = 3.4). EdU incorporation indices had indicated a much lower ratio (0.35/0.26 = 1.35 here or 0.33/0.2 = 1.65 from extensive MARCM lineage studies [18]). The substantial difference is because the length of S-phase, deduced from live imaging, appears to be much greater in layer 2 (838 min) than in layer 1 FSCs (318 min).

A longer S-phase for more anterior FSCs was supported by further live imaging observations. When cells transitioned from G1 they were observed to lose GFP but not to gain RFP

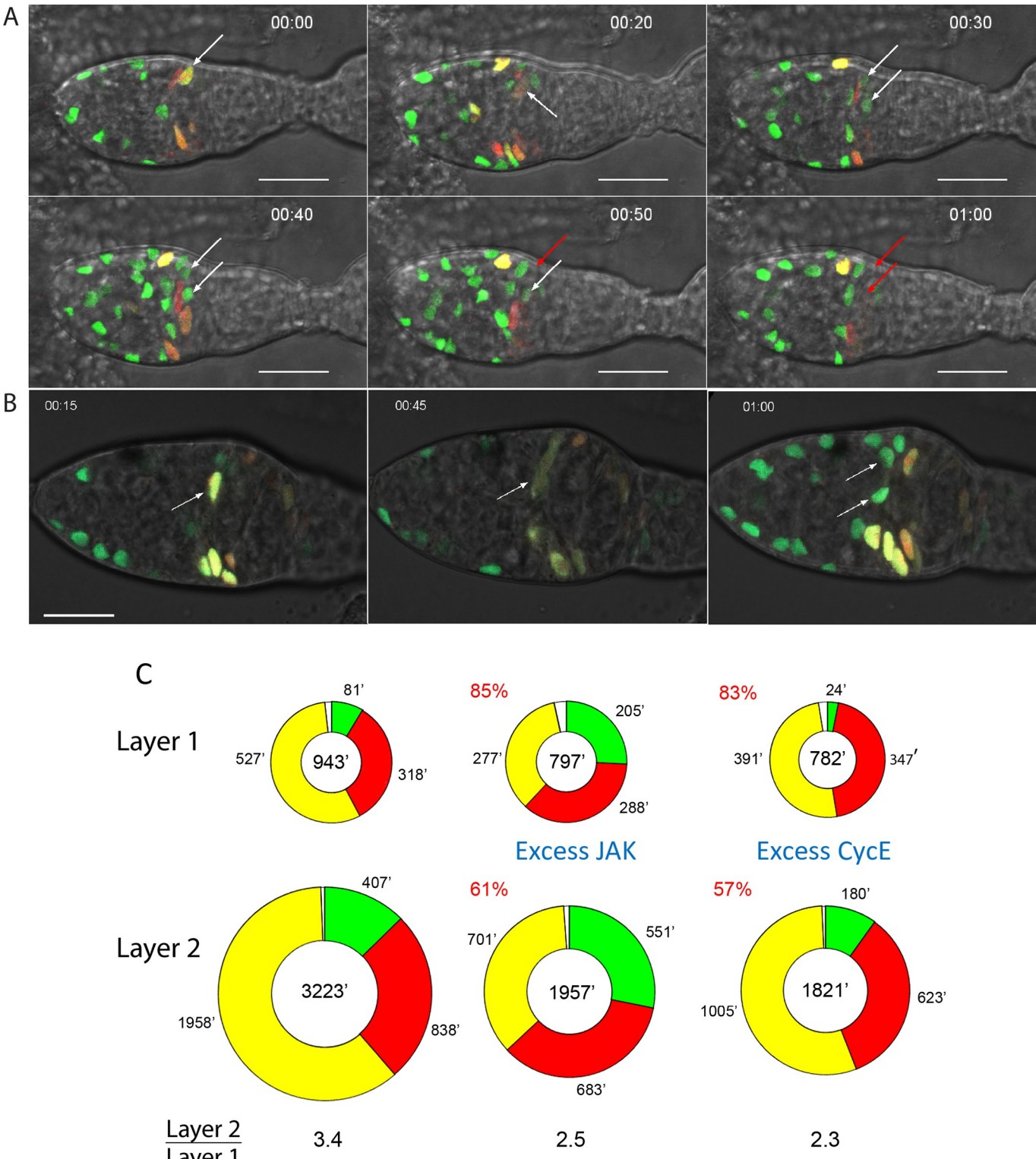

**Fig 4. Live FUCCI reporter imaging shows cell cycle transitions and phase durations.** (A, B) Time-stamped frames from live imaging of (A) control and (B) *C587>Hop* germaria. (A) A yellow G2 cell (white arrow) changes to M-phase morphology at 20min, producing two G1 daughters (white arrows) with only GFP (no RFP) signal at 30min and at 40min. One daughter (red arrow) has lost GFP, indicating S-phase (red arrow) at 50min. The second daughter lost GFP, entering S-phase at 60min. The highlighted cell and its daughters are scored as layer 1 FSCs because they are at the posterior margin of strong *C587-GAL4* expression driving the FUCCI reporter transgenes. See S1 Movie. (B) A yellow G2 cell (white arrow) changes to M-phase morphology at 45min, and then produced two G1

(green) daughters (white arrows) by 1h. The highlighted cells are two diameters away from the posterior edge of *C587-GAL4* expression, indicating that they are layer 3 FSCs. Cell cycle transitions for layer 3 FSCs were observed only for cells with increased JAK-STAT activity, consistent with slow cycling of anterior FSCs resulting partly from insufficient JAK-STAT activity. See S2 Movie. Scale bar is 10μm. (C) Summary of calculated average duration of G1 (green), S (red), G2 (yellow) and M (white) phases of the cell cycle from the sum of all live imaging assays for control (left), *C587>Hop* (middle) and *C587>CycE* (right) germaria. Total average cell cycle length is shown in the center of each circle. For each genotype, the ratio of the whole cell cycle length for layer 2 relative to layer 1 FSCs is shown. The lengths of cell cycles as a percentage of controls are shown in red for excess JAK and CycE. Raw data are in S1 Data (tabs Figs 4, 4-WT and 4-UAS-CycE and 4-UAS-Hop).

within the subsequent observation period (ranging from 15 to 200 min; average 114 min, 22 examples), consistent with a delay before sufficient RFP accumulates to visible levels, as discussed earlier. Thus, early S-phase cells are colorless and late S-phase cells have RFP. From fixed images, EdU labeling shows that layer 1 FSCs were more frequently in S-phase than layer 2 FSCs. Despite this, among 92 layer 1 FSCs, an RFP-only cell was seen for a total of only 204 mins (4 cells), whereas RFP-only layer 2 FSCs (from a smaller sample of 72 cells) were seen for a total of 1344 mins (11 cells), consistent with a considerably longer S-phase.

None of the 20 layer 3 FSCs tracked underwent a FUCCI reporter change, consistent with the expected slow cycling of those cells but providing no quantitative data with regard to cell cycle times.

The deduced cell cycle times for FSCs in different locations can be usefully compared to various quantitative deductions from cell lineage studies. In an extensive study of the cells resulting from marked FSCs over a 3d period, it was ascertained that about four FCs are produced for every EC produced [4]. It has also been shown that 5–6 FCs are produced from FSCs per egg chamber budding cycle of about 12h, that FCs derive directly from layer 1 FSCs, and ECs derive directly from anterior FSCs [4,10]. Production of 5–6 (say 5.5) FCs from 8 layer 1 FSCs (without depletion) in 12h would require an average cell cycle time of 1047 mins (720 mins x 8/5.5). Live imaging studies revealed a cell cycle time of 943 mins. For EC production at ¼ the rate of FCs to be sustained by division of only six layer 2 FSCs (the few FSCs in layer 3 divide much more slowly), those FSCs should have a cell cycle time three-fold (4 x 6/8) greater than for layer 1 FSCs. Live imaging studies deduced a ratio of 3.4.

Thus, the cell cycle durations measured by live imaging support a scenario where FC production is supported by layer 1 FSC divisions and EC production is supported by anterior FSC divisions. It was previously shown that roughly half of FSCs in a given layer will move to a different layer within 3d [10], thereby allowing individual FSC lineages to include both ECs and FCs, as well as providing the possibility of net flow in one direction or another. Our measurements of cell cycle durations suggest that FSCs must move in both directions at roughly equal frequencies. This dynamic equilibrium serves to create a single FSC community from anterior and posteriorly located stem cell groups, each able to support production of immediately neighboring derivatives.

## Cell cycle times in response to increased CycE or increased JAK-STAT signaling

Live imaging and tracking were also performed for germaria where CycE or Hop was overexpressed, using the same genetic conditions as for fixed-sample FUCCI studies, 3d after shifting to the restrictive temperature. For excess CycE, tracking 48 layer 1 FSCs yielded deductions of a G1 phase of just 24 mins, S phase of 347 mins, G2 of 391 mins, and M phase of 20 min, summing to a total cell cycle time that is 83% (782/943) of controls (Fig 4C). Despite the faster overall cycling, S-phase was estimated as slightly extended (347 min vs 318 min for wild-type). Excess CycE was similarly found to shorten G1 but extend S-phase in wing disc cells [40]. From 35 layer 2 FSCs, G1 duration was deduced to be 180 mins, S was 623 mins, G2 was 1005

mins, and M was 13 mins, for a total of 1821 mins, which is (1821/3223) 57% of the control cell cycle time. Thus, excess CycE appears to speed cycling more effectively in layer 2 FSCs than in layer 1 FSCs, consistent with the more prominent G1/S restriction in more anterior wild-type cells.

Excess JAK-STAT resulted in deduced phase durations of G1: 205 mins, S: 288 mins, and G2: 277 mins (M: 27 mins) among 91 layer 1 FSCs for a total cell cycle time of 797 mins, which is 85% (797/943) of controls (Fig 4B and 4C, S2 Movie). For 51 layer 2 FSCs, deduced phase durations were G1: 551 mins, S: 683 mins, G2: 701 mins, and M: 22 mins, for a total cell cycle time of 1957 mins, which is 61% (1957/3223) of controls. The reduction of cell cycle time in layer 1 and layer 2 FSCs resulting from excess JAK-STAT was principally due to a reduction in time spent in G2 (47% reduction in layer 1, 64% reduction in layer 2), with G1 residence significantly increased. This contrasts with the large reduction in average G1 duration observed for excess CycE (70% in layer 1, 56% in layer 2). The faster cycling of FSCs with increased JAK-STAT activity is consistent with the observed amplification of marked FSCs with elevated JAK expression in lineage assays, despite an increased frequency of FSC to FC conversion [18]. Based on our measurement of cell cycle times, the number of new cells produced every 12h when JAK-STAT pathway activity is increased is expected to be 7.2 (number of cells x 720 min period/ cell cycle time: 8 x 720/797) for 8 layer 1 FSCs, compared to 6.1 (8 x 720/943) for wild-type, and 2.2 (6 x 720/1957) for 6 layer 2 FSCs, compared to 1.3 (6 x 720/3223) for wild-type, plus any new FSCs produced from layer 3. Total FSC production per 12h budding cycle is therefore estimated to be at least 9.4 cells compared to 7.4 for controls, an increase of almost 30%.

The fractional decrease in cell cycle time induced by excess JAK was greater for layer 2 (38%) than layer 1 FSCs (15%), consistent with the anterior bias of *C587-GAL4* expression [10] and the higher levels of endogenous JAK-STAT activity in layer 1 FSCs. However, the deduced cell cycle time for layer 1 FSCs (797 mins) was still less than half that of layer 2 FSCs (1957 mins). Under the same conditions of this experiment, we previously showed that JAK-STAT activity was spatially uniform over the FSC domain and that the EdU indices in the two layers were identical (both 33%) [18]. The discrepancy between the two types of measurement is because the length of S-phase remains substantially different between layers 1 and 2 (estimated as 288 min and 683 min, respectively), even when JAK-STAT pathway activity has been equalized. Thus, when JAK-STAT pathway activity is spatially uniform over the FSC domain the less frequent entry but longer passage in layer 2 FSCs happen to balance exactly the more frequent entry and shorter passage through S-phase in layer 1 FSCs to give the same EdU index. Despite a deceptive, shared EdU index, live FUCCI imaging showed that layer 1 and 2 FSCs with the same level of JAK-STAT signaling have very different cycling rates (by about 2.5-fold).

In summary, live imaging of FUCCI reporters has allowed deduction of the absolute timing of different cell cycle phases (Fig 4C). The results showed that the difference in average division rate between posterior and anterior FSCs is much greater than previously estimated from EdU indices (3.4 vs 1.7). Moreover, posterior FSCs still divide much faster than anterior FSCs when JAK-STAT signaling is spatially uniform, indicating major input from other AP-patterned signals. Results also revealed that the length of S-phase was not constant among all FSC locations and genotypes investigated, with the consequence that EdU index cannot reliably be used to infer cell division rates quantitatively. This limitation is most severe when comparing FSCs in different layers because S-phase is much longer for layer 2 (838 min) than layer 1 FSCs (318 min). Comparison of different genotypes for a single FSC layer showed more limited variation (layer1, 347 and 288 vs 318 min: layer 2, 623 and 683 vs 838 min, comparing excess CycE and excess JAK, respectively, to controls).

## FSC division rate tracked by H2B-RFP dilution

We additionally aimed to measure FSC division frequency by using dilution of stable H2B-RFP protein. We mobilized a *UAS-H2B-RFP* P-element insertion and identified a third chromosome insertion with particularly strong expression when driven by *act-GAL4*. In animals also containing a temperature-sensitive *GAL80* transgene, we found that there was little H2B-RFP expression unless animals were moved from 18C to 29C, and we explored various protocols for initiating H2B-RFP expression. Similar results were obtained for any incubation at 29C longer than about 48h, whether during adulthood, pupal stages or throughout post-embryonic development. Although it might be expected that H2B-RFP is synthesized and incorporated into chromatin more efficiently in dividing cells, we observed efficient labeling of quiescent adult ECs even over short time periods. Overall, the patterns of labeling intensities among different cells were very similar to those of GFP from a *UAS-GFP* transgene present in the same animals (S2 Fig). Under all tested conditions, we observed uneven H2B-RFP (and GFP) intensities among germarial cells. Specifically, FC labeling was always weaker than for FSCs and ECs, while r2a EC signals were particularly strong (Figs 5A and S2). We then undertook chase experiments after 4d at 29C, looking for H2B-RFP dilution.

H2B-RFP signal was largely cleared from early FCs to undetectable levels within 7d of chase incubation at 18C (Fig 5B). ECs and anterior FSCs retained a strong H2B-RFP signal, while intensity in some layer 1 FSCs was markedly lower but clearly detectable (Fig 5B and 5C). The intensity of H2B-RFP staining was measured in layer 1 and 2 FSCs, normalizing to layer 3 cells in the same germarium. Prior to chase, average relative H2B-RFP intensities were 1 (layer 3), 0.91 (layer 2), 0.69 (layer 1) and 0.41 (early FCs). After 7d chase at 18C, H2B-RFP intensity was reduced for both layer 1 (0.33; 52% reduction) and layer 2 FSCs (0.56; 39% reduction) relative to layer 3 FSCs, indicating progressively faster division from layer 3 to layer 1. The behavior of cells within a layer varied. We therefore also used a threshold of four-fold dilution to describe how many cells had at least that dilution. After 7d of chase, 45% (5/11) of layer 1 FSCs showed only a weak H2B-RFP signal, below the four-fold dilution threshold, compared to 15% (2/13) of layer 2 and no (0/10) layer 3 cells.

By 12d chase at 18C, the average H2B-RFP signal (normalized to layer 3 FSC levels) had declined to 25% of pre-chase levels in layer 1 and 51% in layer 2. The proportion of cells with greatly diluted signal (more than 4-fold) was 61% (8/13) in layer 1, 17% (2/12) in layer 2 and zero (0/14) in layer 3 (Fig 5D and 5E). By 21d at 18C, even some layer 3 cells had very reduced H2B-RFP signal (Fig 5F). Dilution beyond the four-fold threshold was observed for 18% (4/22), 25% (9/36) and 53% (17/32) of layer 3 cells at 25d, 32d and 34d, respectively. H2B-RFP-depleted layer 2 cells were more frequent at 74% (14/19), 94% (15/16) and 74% (17/23), while almost all layer 1 cells had this property (88% (30/34), 100% (31/31) and 100% (39/39) at these three time-points.

The results of H2B-RFP dilution experiments were consistent with an increasing gradient of division rate from anterior to posterior across the FSC domain. Quantitative inferences were systematically limited because of variability in initial H2B-RFP labeling even among cells in equivalent locations, precluding reliable calculation of the number of cell divisions in a given time period. Differences in H2B-RFP intensity among FSCs in a given layer were more pronounced after long chase periods than after initial labeling. Those variations directly report different life histories for cells assayed in equivalent final positions. They likely have two distinct origins. First, lineage analyses suggest that FSC behaviors are stochastic; this may include significant variation in division rate within a single FSC layer. Second, it is known that FSCs can exchange layers and perhaps also enter EC territory and then return [10].

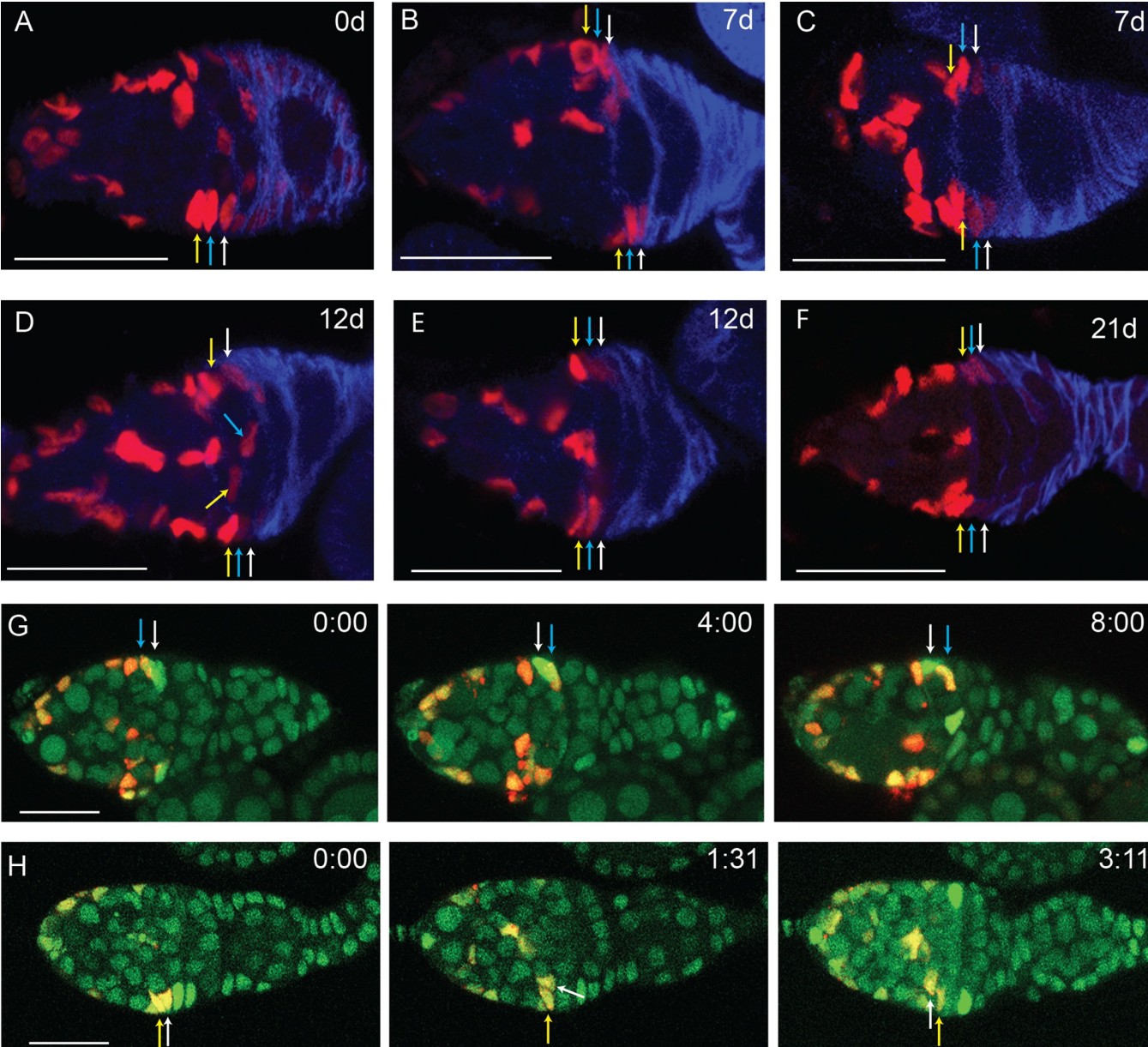

**Fig 5. H2B-RFP dilution shows A/P division gradient and exchange of FSCs between FSC layers.** (A-F) Flies with *UAS-H2B-RFP*, *actin-GAL4* and *tub-tsGAL80* were kept at 29C for 4d and then moved to 18C for the times indicated. Ovaries were stained for Traffic Jam (not shown) and Fas3 (blue). (A) Prior to 18C chase, H2B-RFP showed strong expression in layer 2 and 3 FSCs (blue and yellow arrows) and region 2a ECs, weaker expression in layer 1 FSCs (yellow arrow) and even weaker expression in FCs, which express Fas3 strongly. Three consecutive middle z slices were combined for this image. (B, C) Germaria from flies kept at 18C for 7d had no detectable H2B-RFP in FCs, and variable dilution of H2B-RFP across FSC layers, ranging from minimal for the FSCs indicated by arrows in (B), to strong dilution of FSCs in layer 1 (white arrows in (C)) and layer 2 (bottom blue arrow in (C)). (B) and (C) have 2 and 3 z slices combined, respectively. (D) Germarium from a fly kept for 12d at 18C showing large differences between FSCs in a single layer. Exceptional FSCs have high RFP expression in layer 1 (top white arrow) and weak RFP expression in layer 3 (middle yellow arrow). Layer 2 FSCs (blue arrows) had a mixture of strong and weak expression (blue arrows), and most layer 1 cells had very weak expression (bottom white arrow). 4 middle z slices were combined. (E) Germarium kept at 18C for 12d showing diluted layer 1 and 2 FSCs (top white and blue arrows) and a layer 1 FSC (bottom white arrow). A singe z slice is shown. (F) Germarium kept for 21d at 18C showing strongly reduced H2B-RFP in layers 1–3 FSCs (top) and layers 1–2 (bottom) but one strong layer 3 cell (bottom). A singe z slice is shown. (G) Live imaging frames at 0, 4 and 8h of a germarium kept for 21d at 18C shows a layer 1 cell (white arrow) moving into layer 2, and a layer 2 cell (blue arrow) moving into layer 1. Layers were determined by H2B-RFP expression level, with most layer 1 cells expressing very weak RFP and layers 2 and 3 expressing stronger RFP. See also S3 Movie. (H) Live imaging of a germarium kept at 18C for 16d shows a layer 2 FSC (white arrow) moving to layer 3 at 1h 31min and to region 2a at 3h 11min. A layer 3 FSC is indicated by the yellow arrow for reference. See also S4 Movie. All scale bars, 20μm.

We took advantage of varied H2B-RFP intensities to facilitate tracking cells by live imaging and look directly for FSC movements between layers. We included a *ubi-GFP* marker to visualize all somatic cells and examined germaria that had been chased for at least 7d at 18C. The pattern of H2B-RFP intensities allowed us to define the location of layer 1 and other layers in the absence of Fas3 staining. We observed instances of a layer 2 FSC moving into layer 1, and vice versa (Fig 5G, S3 Movie). Movement in both directions is consistent with our inference that division and differentiation rates are balanced within each layer, and the consequent expectation of no net A/P flow. We also observed a layer 2 FSC moving into layer 3 and then into the location of a region 2a EC (Fig 5H, S4 Movie), and a layer 3 cell moving into layer 1 (S5 Movie).

## MARCM clonal analysis of responses to regulators of FSC cycling

The mechanisms that regulate FSC proliferation can also be probed by clonal analyses, using the MARCM technique [46], where the behavior of GFP-marked FSCs of altered genotype is measured in the context of unmarked normal cells (Fig 6B and 6C). We have undertaken such assays here and in the past, using a standard protocol with measurements at 6d and 12d after clone induction [18]. Division rate is indicated by the EdU index of marked cells in specific locations at 6d. Our live analyses of FUCCI reporters suggests EdU index can be a good qualitative indicator of changes in division rate within a given layer, but may be quantitatively inaccurate because of changes to the length of S-phase. In clonal analyses, we also measure relative rates of conversion of FSCs to ECs and to FCs (at 6d), changes in the A/P distribution of genetically altered marked FSCs (at 12d), and whether the marked FSC population grows or declines in competition with unmarked FSCs (by 12d) [18]. These additional measurements (Fig 6A) allow appraisal of whether factors affecting FSC division rate also affect other FSC behaviors, and whether the net effect on the number of marked FSCs suggests an accord between FSC proliferation and the division rate indicated by EdU index.

For example, in previous studies it was found that increased CycE raised the EdU index, did not substantially affect conversion of FSCs to ECs or FCs, and resulted in amplification of the marked FSC population [18]. By contrast, loss of Wnt signaling did not significantly affect the FSC EdU index but increased conversion of layer 1 FSCs to FCs and caused most FSCs to move posteriorly into layer 1, resulting in a net decline of marked FSCs [10,18]. The effects of altered JAK-STAT signaling were more complex. Loss of STAT drastically reduced the EdU index but loss of marked FSCs was tempered by reduced conversion of FSCs to FCs. Conversely, the increased EdU index in both FSCs and ECs caused by increased JAK-STAT signaling was partially countered by increased conversion of FSCs to FCs, but still led to a large increase in marked FSCs [18].

## JAK-STAT stimulation of EdU index depends on G1/S and G2/M actions

The EdU index of marked FSCs in each layer was increased by expression of excess CycE (46%, 30%, 11%) or Stg (45%, 30%, 13%) compared to controls (33%, 20%, 7%), while CycE and Stg together produced an even larger increase (76%, 70%, 55%), higher than we have observed with any other experimental manipulation (Fig 6D). These results were broadly similar to those from FUCCI analyses. In the MARCM clone tests, transgenes are driven by a combination of *tub-GAL4* and *act-GAL4*, which are expressed relatively evenly among all FSC layers and ECs, while *C587-GAL4*, used in FUCCI tests, is expressed at higher levels in anterior FSCs and ECs than in posterior FSCs. Accordingly, EdU index increases were more pronounced in posterior cells in clonal analyses than in FUCCI tests.

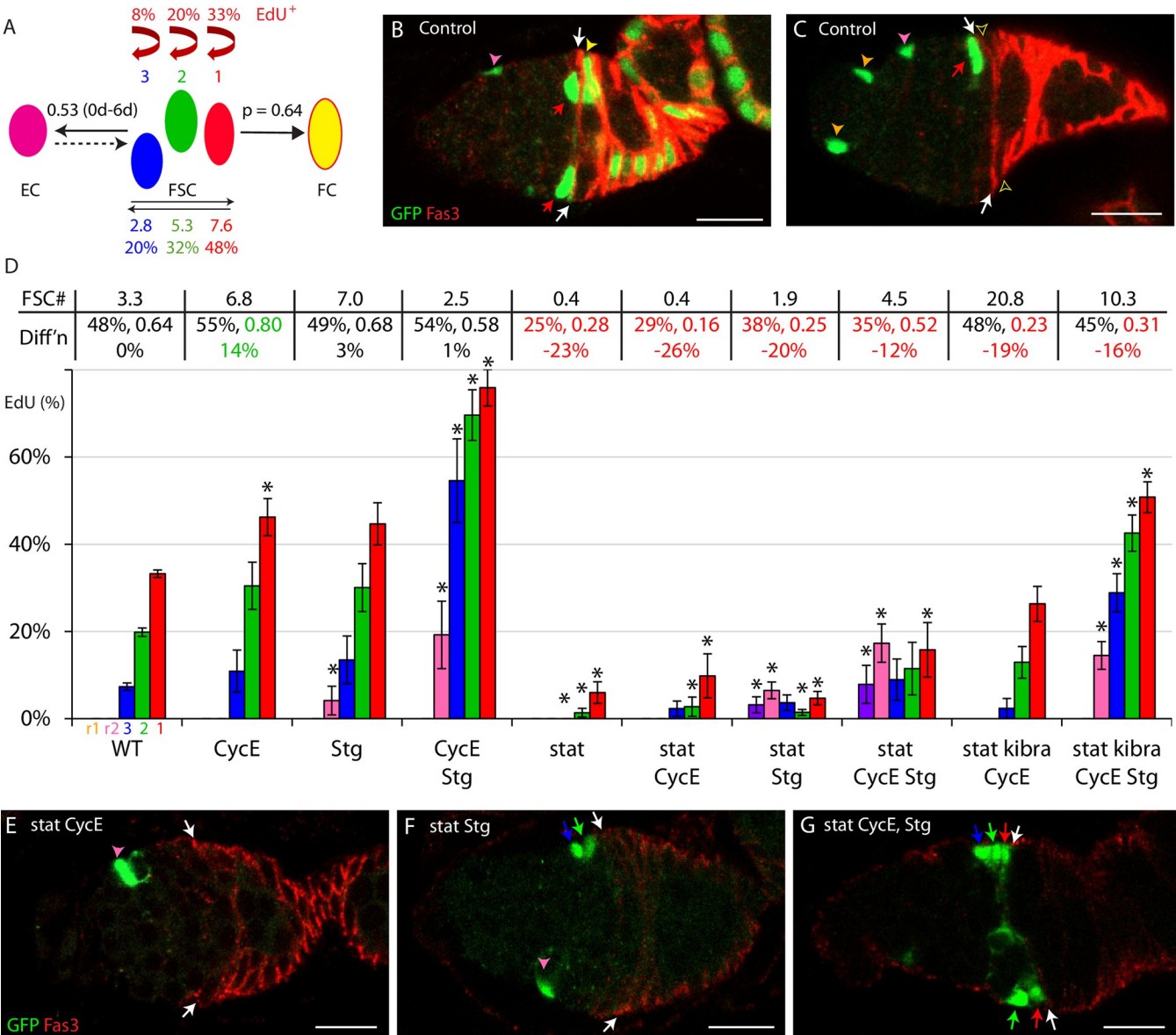

**Fig 6. Restoration of normal division rates to *stat* mutant FSCs requires both excess G1/S and G2/M regulators.** (A) Illustration of separately measured parameters of FSC behavior from multiple control MARCM experiments reported here and previously [18]; EdU indices in each FSC layer, average number and percentage of all FSCs in each layer, ECs produced per anterior FSC from 0-6d, and inferred probability (p) of conversion of a single layer 1 FSC to an FC in each 12h budding cycle. (B, C) Control MARCM samples, illustrating GFP-marked (green) layer 1 FSCs (red arrows), immediate FCs (filled yellow arrowhead in B, none in C; empty arrowhead) just posterior to the anterior border (white arrows) of strong Fas3 (red) staining, r2a (pink arrowheads) and r1 ECs (orange arrowheads, only in C). (D) Percentage of marked cells of the indicated genotypes that incorporated EdU for cells in the location of r1 ECs (purple), r2a ECs (pink), layer 3 (blue), layer 2 (green) and layer 1 (red) FSCs. SEMs and significant differences from control values are indicated (*, p<0.05, n-1 chi-squared test). All values derive from experiments where at least fifty ovarioles were scored. The total number of marked FSCs scored for an altered genotype ranged from 27–199 (average 101) for layer 1 FSCs, 28–139 (average 78) for layer 2 FSCs, and 24–169 (average 56) for layer 3 FSCs; precise values (for ECs also) are in S1 Data. For each genotype, above the graph, the average number of marked FSCs at 12d is indicated ("FSC #"). The two values on the top line of the row labeled "Diff'n" are two additional parameters derived from each MARCM lineage analysis: the percentage of FSCs in layer 1 at 6d, followed by the average probability of a layer 1 FSC becoming an FC in a single 12h budding cycle. The inferred percentage loss of FSCs due to altered FC differentiation per 12h budding cycle was calculated from those two measurements (see Materials and Methods) and is written underneath. (E-G) MARCM samples for *stat* mutant FSCs expressing excess (E) CycE, (F) Stg or (G) both, with the Fas3 (red) border indicated (white arrows). (E) Loss of STAT activity caused greatly reduced division and anterior displacement of marked cells. (G) Excess CycE and Stg together partly restored FSC numbers (red arrow- layer 1; green arrow- layer 2; blue arrow- layer 3), with (F) much less restoration by excess Stg alone. Scale bars 10μm. Raw data are in S1 Data (tabs Fig 6- RawECdata, Fig 6RawFSCdata, Fig 6 EdU, Fig 6 p-values, Fig 6 Diffn).

Among EC derivatives of marked FSCs, there was no response to excess CycE (Fig 6D). Excess Stg triggered some S-phase entry for r2a ECs but addition of excess CycE substantially increased the EdU index (from 4% to 19%), as observed in FUCCI analyses (Fig 3). FSC-derived r1 ECs were not found in S phase even with excess CycE and Stg.

We then tested whether deficiencies in JAK-STAT signaling could be compensated by excess CycE or Stg. As described previously [18], excess CycE only very weakly increased the EdU index for *stat* mutant FSCs (9.8%, 2.8%, 2.3% vs 6.0%, 1.4%, 0% for layers 1, 2 and 3, respectively) (Fig 6D and 6E). Excess Stg produced a similar result (4.7%, 1.5%, 3.7%) (Fig 6D and 6F). However, EdU incorporation was significantly increased for *stat* mutant FSCs, especially in more anterior locations, by providing excess CycE and Stg together (15.8%, 11.5%, 9.0%) (Fig 6D and 6G).

This synergy was tested further in the background of an additional genetic alteration. Yki activation through the Hippo pathway (using a *kibra* mutation) was described previously to enhance EdU incorporation in *stat* mutant FSCs alone (6.5%, 4.7%, 0.7% for layers 1–3), and more potently together with excess CycE (26.3%, 12.9%, 2.4%) [18]. There is some evidence that Yki acts in FSCs through increasing transcriptional induction of *cycE* [17]. Addition of both *UAS-Stg* and *UAS-CycE* to *kibra stat* mutant FSCs increased EdU incorporation to levels significantly higher than for wild-type cells (51%, 43%, 29% vs 33%, 20%, 7%) (Fig 6D). Thus, the normal input from JAK-STAT signaling was consistently compensated by stimulation of both G1/S and G2/M transitions but only poorly by reagents expected to act on only one phase transition. Notably, even though the normally graded input from posterior (high) to anterior (low) of JAK-STAT was replaced by excess CycE and Stg driven by a constitutive promoter (together with uniform Hippo pathway inactivation when *kibra* was altered) in these experiments, the EdU index of posterior FSCs remained higher than that of anterior FSCs in all cases, indicating strong A/P-graded influences other than JAK-STAT.

EdU incorporation into r2a ECs was not stimulated by excess Stg when *stat* activity was absent. However, *stat* mutant ECs did incorporate EdU when both *UAS-Stg* and *UAS-CycE* were provided in the presence (10%) or absence of a *kibra* mutation (14%), including some r1 ECs (8%) in the latter case (Fig 6D). Thus, the low JAK-STAT activity normally present in r2a ECs is necessary for the limited entry of those cells into S-phase stimulated by excess Stg, and likely acts by facilitating the G1/S transition.

All of the above results are consistent with the conclusions from FUCCI analyses that JAK-STAT signaling normally stimulates both G1/S and G2/M transitions over the EC and FSC domains, provided that EdU indices report relative cell division rates with reasonable fidelity. The loss or gain of FSCs over the 12d of a MARCM experiment provides a further check. If a specific genetic manipulation has no effect on FSC location or conversion to ECs or FCs, we would expect a straightforward correlation between the number of marked FSCs accumulating and the division rate of the most proliferative (layer 1) FSCs. That correlation could, however, be disrupted most potently by a significant change in the frequency of FSC to FC conversion. We therefore report for each genotype the number of marked FSCs at 12d (top line Fig 6D), the percentage of FSCs in layer 1 at 6d (normally 48%, Fig 6A) and the probability of a single layer 1 FSC becoming a FC in one cycle of egg chamber budding (p = 0.64 normally, calculated from 6d data, Fig 6A) as measures of FC production ("Diff'n") [18]. Below the latter two values is the calculated expected percentage change in FSC loss due to FC formation relative to wild-type per cycle of egg chamber budding (a positive number, which can result from a higher fraction of FSCs in layer 1 or a greater rate of conversion of layer 1 FSCs to FCs, indicates greater loss; see Materials and Methods).

Excess CycE or excess Stg alone did not greatly alter FSC conversion to FCs. Each increased the accumulation of marked FSCs in accord with the observed increases in layer 1 FSC indices.

Both factors together, however, did not increase FSC numbers, despite the very high EdU index and no significant measured effect on FSC to FC conversion. Since FSC accumulation is lower than with either excess CycE or Stg alone, it is likely that excessive artificial stimulation of cell cycle transitions has led to a replicative crisis, involving some prolonged S-phases or perhaps cell death.

In the absence of functional STAT, there is reduced FC differentiation [18]. That property was substantially retained with excess CycE, Stg, or both, with calculated reductions of FSC to FC conversion frequencies ranging from 12–23% (Fig 6D). Consequently, we would expect that the number of marked FSCs for all of these genotypes will be higher than expected for a given FSC division rate. Nevertheless, the accumulation of FSCs should correlate with the relative EdU indices among these *stat* mutant genotypes if EdU indices are reporting division rates reasonably accurately. Indeed, provision of excess CycE and Stg together did increase *stat* mutant FSC numbers by more than either alone (4.5 vs 0.4 and 1.9), as seen for EdU indices, supporting the inference that FSC division is enhanced by complementing both a G1/S and G2/M transition defect due to the absence of functional STAT (Fig 6D–6F). In the additional presence of a *kibra* mutation, the addition of excess CycE alone already leads to saturating numbers of marked FSCs, so it is not possible to measure any further increases. The number of marked FSCs remains very high when excess Stg is added, suggesting that the inferred replicative crisis or other stresses encountered by FSCs expressing excess CycE and Stg are largely avoided if the magnitude of all input provoking cell cycle transitions is lessened by elimination of JAK-STAT signaling (Fig 6D).

## Discussion

The regulation of adult stem cell proliferation is universally important to maintain stem cell populations while providing an adequate supply of derivatives to maintain tissues throughout life. For paradigms, like FSCs and mouse gut stem cells, where stem cell division and differentiation are independent, division and differentiation rates must be balanced over the whole stem cell population and mutations that accelerate division may often be critical to seeding cancers [4,5]. Additionally, for FSCs, there is pronounced spatial regulation of division rates. Here, we have introduced FUCCI cell cycle reporters to gain new insights into the regulation of FSC division rate.

### Insights from FUCCI reporters: Absolute cell cycle times

Live imaging with FUCCI reporters allowed us to record the frequency of cell-cycle transitions and deduce absolute cell cycle times. These parameters have not previously been measured for FSCs and are generally not known in other adult stem cell paradigms. A recent study of Drosophila female GSCs is a notable exception [47]. In standard long-term lineage experiments, the total number of divisions of an FSC cannot be measured precisely because FC proliferation obscures the exact number of FCs derived directly from an FSC. Over a short period of time (like 3d), the number of FCs produced and FSC division rates can be estimated [4]. However, that does not reveal the locations of FSC divisions and hence any information about the spatial pattern of FSC divisions. Here we found that, on average, layer 1 FSCs divide 3.4-fold faster than their anterior neighbors, roughly twice the difference previously estimated from EdU indices (Fig 7). The discrepancy arises because S-phase is significantly longer in anterior than posterior FSCs ("deceptively" elevating anterior FSC EdU indices for a given rate of cycling).

The dramatic cycling differential between layer 1 and layer 2 FSCs, and the estimated average absolute cell cycle durations (943 min and 3223 min, respectively) revealed by live FUCCI imaging have important implications. First, the number of new cells deduced to result from

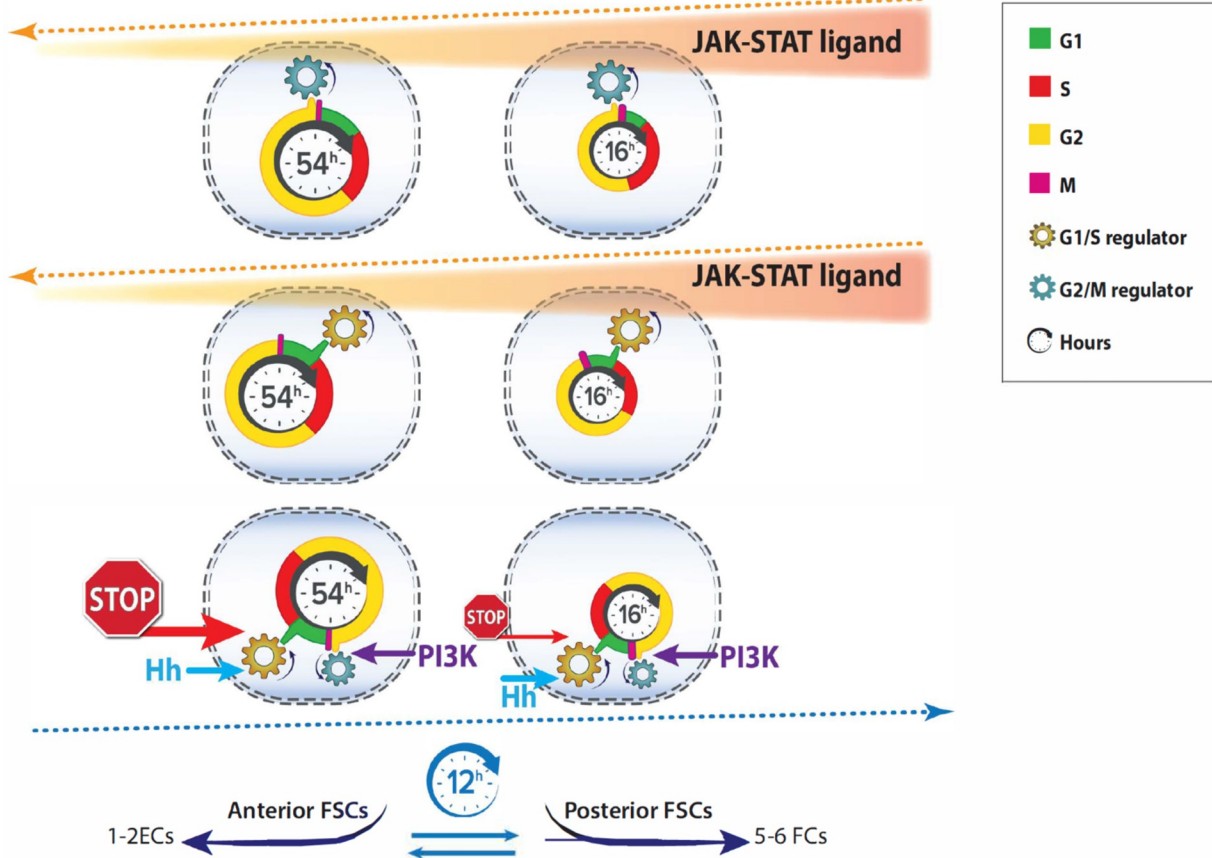

**Fig 7. Regulation of FSC cell cycles and implications for stem cell flux.** The cartoon summary depicts three of the eight most posterior, layer 1 FSCs (right) and three of their six anterior layer 2 neighbors (left). Cells in the same layer experience the same signals and responses but each image highlights different responses. Cell cycle duration was measured in this study by live imaging with FUCCI reporters. It was much shorter (3.4-fold) for posterior FSCs (roughly 16h) than for anterior FSCs (roughly 54h). G1 (green) is very short in posterior FSCs. G1 was progressively longer in more anterior locations, culminating in anterior ECs (not shown), which were almost always in G1. This suggests a strong G1/S restriction in the anterior of the germarium that is gradually reduced towards the posterior. JAK-STAT signaling declines from posterior to anterior (red/orange shading) and was found to promote both G2/M transitions (blue gear; top row) and G1/S transitions (gold gear; second row), partly accounting for faster cycling of posterior FSCs. However, when JAK-STAT signaling was uniform, posterior FSCs still cycled much faster (2.5-fold) than anterior FSCs, indicating the presence of at least one other major spatially restricted cell cycle signal. Hh ligand (cyan arrows, third row) emanates from cells anterior (left) to FSCs and acts principally to stimulate G1/S transitions (gold gears) [17]. PI3K pathway activity (purple arrows) principally affects the G2/M transition (blue gears); the spatial origin or modulation of signals activating the PI3K pathway is not known. We speculate that the missing spatial signal may primarily restrict G1/S passage in more anterior cells (red "STOP" signal). The cell cycle wheel is rotated in row 3 for the diagrammatic convenience of showing all inputs clearly. The net result of the 3.4-fold faster cycling of posterior FSCs and the measured 16h cell cycle is that in each 12h egg chamber budding cycle, division of eight layer 1 FSCs suffices to produce the observed number of additional cells that become FCs (five to six), while anterior FSCs produce the previously measured output of ECs (one fourth that of FCs) [4]. FSCs were known to exchange layers but it was not known if this occurred principally or only in one direction [10]. Quantitative cell cycle measurements now predict an equal flow in each direction (paired reciprocal blue arrows). Live imaging of FSCs, aided by H2B-RFP labeling in this study, directly showed FSCs moving from layer 1 to layer 2, and *vice versa*.

division of eight layer 1 FSCs in each 12h egg chamber budding cycle (8 x 720/943 = 6.1) matches the measured number of founder FCs (5 to 6) for each egg chamber [4,10], while division of six layer 2 FSCs are deduced to produce new cells at a roughly four-fold lower frequency (6/8 x 943/3223 = 0.22), matching new EC output [4,10]. Earlier studies indicated that layer 1 FSCs directly become FCs and anterior FSCs (layer 2 or 3) directly become ECs, but that FSCs can exchange A/P locations, forming a single community of stem cells supporting lifelong EC and FC production [4,10,18]. We can now deduce from cell division rate measurements that the exchange between layers 1 and 2 is roughly at equilibrium, with little or no net

flow from anterior to posterior within the FSC domain and therefore no systematic difference in the projected longevity of anterior versus posterior FSCs (Fig 7). We also used H2B-RFP labeling to observe movement of FSCs in both directions, for the first time, in this study.

Second, our measurement of FSC division rates also provides further confirmation of the current model of FSC behavioral dynamics. We found that an estimated total of 7.4 cells are produced from layer 1 (6.1 cells) and layer 2 (1.3 cells) per 12h egg chamber budding cycle. This is a close match to the number of new FCs (5.6 cells) and ECs (1.4 cells) produced, inferred from lineage analyses [4]. The original postulate of two FSCs, each producing one FC per egg chamber budding cycle, is incompatible with our direct observations and measurements of FSC cycling, as are any minor variants of the original model that do not take into account all FC-producing cells [2].

Live imaging measurements also resolved a conundrum with regard to spatial regulators of FSC division. It was previously found that JAK-STAT signaling is a major contributor to FSC division in all locations but the FSC EdU index remained spatially graded in the absence of STAT activity, clearly indicating another source of A/P pattern [18]. By contrast, supplying excess JAK-STAT activity in a pattern complementary to the normal posterior to anterior gradient, thereby equalizing activity over the FSC domain, resulted in almost identical EdU indices among the three FSC layers [18]. Here we found that under those conditions, layer 1 and layer 2 FSC FUCCI profiles remained very different. Moreover, the overall cell cycle duration ratio (layer 2: layer 1) declined, but only from 3.4 to 2.5. EdU indices were only co-incidentally equal among FSCs in different layers because S-phase remained substantially longer in anterior than posterior FSCs, and therefore provided a deceptive measure of FSC cycling rates. Thus, FUCCI reporter tests now clearly show a major spatial influence on FSC cycling under conditions of both STAT absence and uniform JAK-STAT activity. This unknown second influence supplements the spatial consequences of graded JAK-STAT signaling on cell division rate under normal conditions (Fig 7).

The method used to derive estimates of cell cycle phase durations uses raw observations of partial cell cycles from multiple cells of the same type (for example, layer 1 FSCs). This approach has not been commonly used in the past to our knowledge. The method does depend on sufficient sampling of cells and is partially compromised if the cell population studied has extensive cycling heterogeneity (see S1 Methods: "Live FUCCI Calculation Method" and "Sampling Considerations for Live FUCCI Calculation Method"). However, similar limitations apply to following whole cell cycles, where failure to score complete cell cycles for all observed cells may severely bias assessment of average cell cycle parameters in favor of the faster cycling cells. A major advantage of tracking only partial cell cycles from a large cell sample is that the tracking period can be short. In some cases, longer-term tracking may not be possible and in all cases, a shorter tracking period is less likely to alter normal physiological behaviors, whether distortions are potentially induced by placing the tissue in a different environment for observations or by the observation procedures themselves. Based on our experience with FSCs, we suggest that the same approach could usefully be extended to numerous other stem cell and other paradigms.

## H2B-RFP dilution: Further evidence of graded division and visualization of FSC A/P exchanges

Results of H2B-RFP dilution experiments supported the conclusion of faster cycling of layer 1 than layer 2 FSCs, with very little cell division further anterior. This approach had some systematic limitations of uneven cell labeling prior to chase periods that precluded accurate measurement of division rates of cells. The method also cannot report the division rate of a cell in a

specific location because FSCs can change their location over time. For example, an FSC scored in layer 2 after several days of chase may have resided in layer 1 or layer 3 for some of that time. Indeed, there were often dramatic differences of H2B-RFP signal among FSCs in a given layer after a chase period, providing valuable insights into the variable history of seemingly equivalent cells, whether due to changes in cell location or stochastic differences in cell division. We also used live imaging of samples after significant periods of H2B-RFP dilution to visualize cell movements directly. Differing H2B-RFP intensities allowed us to define FSC layers and easily track individual cells. This showed for the first time that FSCs can move in either direction between A/P layers, consistent with our deduction from newly-measured cell cycle times that there is little net flow in either direction within the FSC domain (Fig 7).

## JAK-STAT promotes G1/S and G2/M transitions

We also gained some information from FUCCI reporters about specific cell cycle transitions spatially regulated by signals. Reduced JAK-STAT activity in layer 1 FSCs reduced S-phase representation and increased G2-phase frequency, suggesting that JAK-STAT normally promotes both G1/S and G2/M transitions. This deduction is consistent with the effects of increased JAK-STAT activity in each FSC layer. Here, G2 frequency was reduced, clearly indicating G2-M stimulation. The G1:S ratio actually increased, but we still infer facilitation of a G1/S transition by comparing to the effects of excess Stg (expected only to directly stimulate G2-M passage), which results in much higher G1, and lower S-phase frequencies. Promotion of a G1/S transition was also evident for excess JAK-STAT in the most anterior ECs, which normally reside in G1. Similar logic suggested that the PI3K pathway may slightly promote the G1/S transition in FSCs but principally stimulates G2/M transitions in both FSCs and ECs (Fig 7). Strong inhibition of Wnt signaling was largely without effect on EC or FSC cell cycle phase occupancy when tested alone or in conjunction with excess JAK-STAT pathway activity.

The deduction that JAK-STAT signaling acts on both G1/S and G2/M transitions was supported by MARCM lineage studies, which measured cell division indirectly through EdU incorporation indices and the accumulation of marked FSCs. The greatly reduced division of FSCs lacking STAT activity was only marginally increased by expression of excess G1/S (CycE) or G2/M (Stg) regulators, but was substantially increased by providing both. In all of these conditions of STAT inactivation, layer 1 FSC division was faster than for anterior FSCs, providing clear evidence of a spatially graded regulator other than JAK-STAT ligand.

The deduction of primary actions of regulators through FUCCI reporter responses benefits greatly from live imaging. Excess CycE is expected to accelerate the G1/S transition. The G1 proportion in fixed samples and G1 duration in live imaging were indeed both reduced. However, live imaging revealed that G2 was also shortened, while S-phase was lengthened in layer 1 FSCs, neither of which could be deduced just from fixed samples. Similarly, only live imaging revealed that there is strong spatial regulation of the duration of S-phase in wild-type FSCs. These newly-observed phenomena await molecular explanations. Since phases are cyclical, with potentially complex internal sequences of events, and prior phases prepare molecular conditions for future events [34,48–50], it is possible that the spatial variation in S-phase relates to differences in preparation rather than signaling inputs that act during S-phase. Similarly, the effects of excess CycE on S-phase and G2 duration may derive from altered or incomplete events during the greatly accelerated G1 phase.

JAK-STAT signaling is highest in posterior cells. A genetic manipulation (*C587>JAK*) that increased JAK-STAT activity to produce spatially uniform levels over the germarium reduced the discrepancy between layer 1 and 2 FSC cycling periods (from a ratio of 3.4 to 2.5) and promoted some cycling of ECs. Hence, it seems that A/P-graded JAK-STAT signaling is one factor

underlying the patterns of cell division in the germarium. The identity and mechanism of action of other spatial regulators of FSC division are not presently known. In germaria with near-uniform JAK-STAT signaling, the duration of each major cell cycle phase remained much greater in layer 2 than layer 1 FSCs (G1, 551 vs 205 mins; G2, 701 vs 277 min; S, 683 vs 288 min), while passage out of G1 is clearly the biggest limitation for ECs under these conditions. Additionally, in wild-type FSCs there is a strong gradient of G1/S restriction, with 99% of region 1 ECs in G1 compared to just 8% of layer 1 FSCs. The simplest hypothesis is that there is an unknown regulatory signal that primarily restricts G1/S passage in ECs and anterior FSCs (Fig 7). Although we favor this concept of complementary distributions of stimulatory (JAK-STAT) and inhibitory signals, the missing signal could alternatively be stimulatory and posteriorly-biased.

## Comparison to other stem cell paradigms

Mouse gut and epidermal stem cells share with FSCs the characteristics of generally high constitutive division rates and division-independent differentiation [5,6]. Despite the acknowledgment that division and differentiation can be regulated independently, the majority of studies assessing contributions of specific signals in those paradigms measure stem cell survival or amplification, rather than the individual contributing parameters that we have been able to measure for FSCs. Nevertheless, there is evidence of a gradient of Wnt pathway activity in gut crypts [51] and increased division rates were evident in organoids when Wnt pathway activity was increased by *Apc* mutation [52]. Stimulation of gut stem cell division through the EGFR family was also inferred from increased phospho-histone incorporation when the Lrig1 negative feedback regulator was removed [53], while an effect of Notch signaling was inferred from effects on expression of Cdk inhibitors [54]. A positive role of Yki orthologs YAP and TAZ, and antagonism by the Hippo pathway was deduced from EdU incorporation and Ki67 proliferation marker staining [55]. Similar methods were used to deduce that loss of the mitochondrial pyruvate carrier stimulates stem cell division [56]. None of these studies, however, includes a quantitative measure of stem cell cycling rates *in vivo*, and deductions largely rely on uncertain assumptions of constant S-phase length or Ki67 marker interpretation [57]. Since mouse gut crypts can be imaged over long periods of time *in situ* [5], these limitations might be resolved by using live FUCCI reporter imaging in the same way we employed for FSCs.

In the male Drosophila gonad, the cell type most analogous to female FSCs are somatic Cyst stem cells (CysSCs). They share the properties of maintenance by population asymmetry and being more proliferative than immediate neighbors, hub cells [58]. However, their differentiated daughters, in contrast to FCs, do not divide and their overall function guiding germline differentiation combines EC and FC functions. Both Hh and JAK-STAT signaling promote CysSC division, apparently through induction of Hippo pathway components [59]. Other studies suggest that loss of upstream Hippo pathway regulators, Merlin and Expanded may promote CysSC division via increased MAPK and PI3K pathway activities [60]. CysSC cycling was also found to be stimulated by activin receptor signaling, with the presence of Follistatin preventing an analogous response in hub cells [61]. None of these studies measures changes in the rate of CysSC division or exactly how they affect the cell cycle.

Other stem cells, including neural and muscle stem cells, have been studied in the notably different context of transitioning into or out of quiescent states [23,62,63]. Such studies are beginning to reveal a diversity of mechanisms, including clear evidence that the G2/M transition can be regulated, supplementing the long-held assumption that regulation was primarily through entering a G0 state prior to S-phase [24,64]. Our considerable knowledge of relevant

FSC signals, the potential to test causal relationships genetically and to measure cell cycling quantitatively through FUCCI reporters and live imaging, as demonstrated in this work, make FSCs an especially promising paradigm for understanding the regulation of stem cell division rates.

## Materials and methods

### FUCCI reporter experiments

Nuclear-targeted GFP and RFP UAS-driven FUCCI reporters on the second (BL-55121) and third (BL-55122) chromosomes were combined and expressed conditionally using *C587-GAL4* and a second chromosome temperature-sensitive *GAL80* transgene (from BL-7108). 1-3d old female flies of the genotype *C587-GAL4; UAS-FUCCI/tub-ts-GAL80, FRT42D tub-lacZ; (UAS-X)/UAS-FUCCI* were collected, where *UAS-X* was absent (wild-type), *UAS-CycE*, *UAS-Stg* (BL-4778), *UAS-Stg + UAS-CycE*, *UAS-Hop³ᵂ*, *UAS-Hop³ᵂ + UAS-Dap*, *UAS-dnTCF* (BL-4785), *UAS-dnTCF + UAS-Hop*, *UAS-stat RNAi* (BL-31317) + *UAS-DIAP1*, *UAS-PI3K92E A2860C* (aka Dp110-DN) (BL-8289) or *UAS-PI3K92E* (BL-8287) (transgene origins not specified here were the same as in [18]). Flies were incubated at 29C for 3d to inactivate GAL80. Dissected ovaries underwent the EdU and immunohistochemistry protocols described below, without antibody staining for GFP.

### Live imaging

Imaging chambers were fabricated as described previously [10,45]. Ovaries from flies with *C587-GAL4*, *UAS-FUCCI* genotypes described above and incubated at 29C for 3d were dissected into imaging medium formulated as in [65] (20% FBS in Schneiders insect medium, 0.2 mg/mL insulin, penicillin and streptomycin). After separating ovaries into individual ovarioles in imaging medium, 135 μL medium containing ovarioles was mixed with 15 μL Matrigel (Corning), added to the imaging chamber and left covered for 15 min to gel. Wells were then filled to the top with imaging medium. Germaria were generally imaged every 10–20 min but occasionally up to 45 min on a Zeiss LSM 700 confocal microscope. For H2B-RFP live imaging, germaria were imaged every 5 min.

### MARCM clonal analysis

1-3d old adult *Drosophila melanogaster* females with the appropriate genotypes were given a single 30 min (for *FRT40A*) or 45 min (for *FRT82B*) heat shock at 37C. Afterwards, flies were incubated at 25C, with frequent passage on normal rich food supplemented by fresh wet yeast during the 12d experimental period. Flies were dissected at 6d and 12d. Immediately after dissection, 6d ovaries underwent 1h of EdU labelling based on the protocol of the Click-iT Plus EdU Cell Proliferation Kit for Imaging (Invitrogen). Both 6d and 12d ovaries were stained for Fasciclin III (Fas3) and GFP. Ovaries were then manually separated into constituent ovarioles, and mounted using DAPI Fluoromount-G (SouthernBiotech) to stain nuclei. Ovarioles were imaged with a Zeiss LSM700 or LSM800 confocal microscope, operated in part by the Zeiss ZEN software. The entire germarium was captured in the images, as well as an average of 3–4 egg chambers. Collected images were saved as CZI files, and were later analyzed utilizing the ZEN Lite software. We aimed to image at least 50 germaria for every genotype in each experiment.

### MARCM genotypes

Flies with alleles on an *FRT40A* or *FRT82B* chromosome were used in MARCM experiments using the following genotypes:

FRT40A: *yw hs-Flp, UAS-nGFP, tub-GAL4 /yw; act-GAL80 FRT40A / FRT40A; act>CD2>GAL4/ Y*–where Y was +, *UAS-CycE, UAS-Stg,* or *UAS-CycE + UAS-Stg.*

FRT82B: *yw hs-Flp, UAS-nGFP, tub-GAL4 /yw; act>CD2>GAL4 UAS-GFP / Y; FRT82B tub-GAL80/FRT82B (X)*–where X, Y combinations included: (X)–*NM* (control), $stat^{85C9}$, $kibra^{32}$, $wts^{x1}$, $kibra^{32} + stat^{85C9}$, $UAS-CycE + stat^{85C9}$, $UAS-CycE + kibra^{32} + stat^{85C9}$, and (Y) *UAS-Stg.*

## EdU protocol

Ovaries were directly dissected into a solution of 15 µM EdU in Schneider's *Drosophila* media (500µl, Gibco) and incubated for one hour at room temperature. These tubes were laid on their side and rocked manually, to ensure all dissected ovaries were fully submerged. Ovaries were then fixed in 3.7% paraformaldehyde in PBS for 10 minutes, treated with Triton in PBS (500 µl, 0.5% v/v) for 20 minutes, and rinsed 2x with bovine serum albumin (BSA) in PBS (500 µl, 3% w/v) for 5 minutes each rinse. Ovaries were exposed to the Click-iT Plus reaction cocktail (500 µl) for EdU visualization, for 45 minutes. The reaction cocktail was freshly prepared prior to use, with reagents from the Invitrogen Click-iT Plus EdU Cell Proliferation Kit for Imaging, including the Alexa Fluor 594 dye. Ovaries were then rinsed 3x with BSA in PBS (500 µl, 3% w/v) for 5 minutes each rinse.

## Immunohistochemistry

For experiments without EdU, ovaries were dissected directly into a fixation solution of 4% paraformaldehyde in PBS for 10 min at room temperature, rinsed 3x in PBS, and blocked in 10% normal goat serum (NGS) (Jackson ImmunoResearch Laboratories) in PBS with 0.1% Triton and 0.05% Tween-20 (PBST) for 1 h. Monoclonal antibodies for Fas3 were obtained from the Developmental Studies Hybridoma Bank, created by the NICHD of the NIH and maintained at The University of Iowa, Department of Biology, Iowa City, IA 52242. 7G10 anti-Fasciclin III was deposited to the DSHB by Goodman, C. and was used at 1:250 in PBST. Other primary antibodies used were anti-GFP (A6455, Molecular Probes) at 1:1000 in PBST. Ovaries were incubated in primary antibodies overnight, rinsed three times in PBST, and incubated 1–2 h in secondary antibodies Alexa-488 and Alexa-647 (ThermoFisher) at 1:1000 in PBST to label GFP and Fas3, respectively. DAPI-Fluoromount-G (Southern Biotech) was used to mount ovaries.

## Imaging and scoring

All germaria were imaged in three dimensions on an LSM700 or LSM800 confocal laser scanning microscope (Zeiss) and using a 63x 1.4 N.A. lens. Zeiss ZEN software was used to operate the microscope and view images. Images were typically 700x700 pixels with a bit depth of 12. The scaling per pixel was 0.21 µm x 0.21 µm x 2.5 µm. The range indicator in ZEN was used to determine the appropriate laser intensity and gain. ZEN was used to linearly adjust channel intensity for dim signals to improve brightness without photobleaching samples. Images were saved as CZI files and scored directly in ZEN. DAPI and Fas3 staining were used as landmarks to guide scoring. Marked cells were considered FSCs if they were within three cell diameters anterior of the Fas3 border. Cells immediately adjacent to the border were considered to be in Layer 1, with Layers 2 and 3 in sequentially anterior positions. Anterior to the FSC niche, the EC region was roughly divided into two halves, with region 2a ECs immediately anterior to FSCs and region 1 ECs anterior to that. Germaria were also scored (Y/N) for the presence of marked FCs. For the "Immediate FC Method" [18], the presence of an FC immediately posterior to Layer 1 was also scored Y/N. For publication, images were digitally zoomed in ZEN and exported as *tif* files using the "Contents of Image Window" function. Images were rotated in Abode Photoshop CS5 to uniformly orient the germaria.

## Determination of FSC locations and layers

The following protocol is used to assign FSC locations. A variety of indicators help to ensure reproducible determination of the anterior border of strong Fas3 staining. These include germarium width, identifying the youngest stage 2b cyst and noting cell process locations if they are outlined. Starting with a mid-z-section, identify the youngest stage 2b cyst by the criteria of spanning the germarium and not being at all rounded. Most germaria have only a single candidate 2b cyst with a clearly more rounded stage 3 cyst more posterior. In such germaria, the Fas3 border runs along the posterior surface of the 2b cyst. In other germaria (up to about a quarter), a new 2b cyst has formed as a more posterior 2b cyst just starts to round. Here, the strong Fas3 border lies between the two 2b cysts. The strong Fas3 border can then be followed as a continuous surface through neighboring z-sections. Layer 1 cells immediately anterior to the Fas3 border have strong Fas3 staining on their posterior surface but weak or incomplete outlining of the anterior surface by Fas3. Labeling in MARCM clones usually allows visualization of FSC processes (even faintly when using a nuclear-targeted GFP marker). Layer 1 cell processes are present along the Fas3 border, whereas layer 2 processes are anterior to the stage 2b cyst. The widest part of the germarium is generally very close to the Fas3 border, with layer 1 FSCs at the widest location in about three-quarters of samples and immediate FCs at that location in the rest. When examining FUCCI-labeled germaria it is important to be able to distinguish DAPI-stained somatic nuclei without any RFP or GFP signal (S-phase cells) from germline cyst nuclei, which will never express GFP or RFP. This is achieved by recognizing germline cyst nuclei as being larger, more rounded and clustered in known groupings. Specifically, stage 2b cysts are lens-shaped, while 2a cysts are rounded; both contain 16 nuclei. In challenging cases, each germline nucleus can be marked with a central small dot on images showing successive z-sections, so that the 16-cell unit can be outlined, as shown in S1 Fig.

## H2B-RFP dilution experiments

The original third chromosome P-element insertion of mRFP N-terminally tagged H2B as *UAS-H2B-RFP* in a mini[$w^+$] vector is described in [66]. Transposon mobilization, as in [67], was used to identify new insertion sites based on altered eye color and H2B-RFP expression of new lines was assessed after crossing to an *act>GAL4* driver. Parent flies of *genotypes yw hs-flp; tub-GAL80(ts) FRT42D tub-lacZ / CyO; act>GAL4, UAS-GFP / TM2* and *yw; FRT42D / CyO; UAS-H2B-RFP/ TM2* were crossed at 18C to produce female progeny of the experimental genotype *yw hs-flp / yw; tub-GAL80(ts) FRT42D tub-lacZ / FRT42D; act>GAL4, UAS-GFP / UAS-H2B-RFP*. Crosses or adult progeny were shifted to 29C for various times to allow H2B-RFP expression and then back to 18C to prevent further H2B-RFP expression. Flies were regularly changed to new vials with added fresh moist yeast. Ovaries were dissected and stained for Fas3 and Traffic Jam (using guinea pig antibody from Dr. Dorothea Godt). Traffic Jam staining was used to outline nuclei using the Draw Spline Contour function for measuring H2B-RFP intensity with Zen software. For live imaging the genotype of flies was *yw hs-flp; ubi-NLS-GFP FRT40A / tub-GAL80(ts) FRT42D tub-lacZ; act>GAL4, UAS-GFP / UAS-H2B-RFP* so that all nuclei were marked by GFP. Live imaging was performed as described above but with 5 min intervals to track cell movements.

## Statistics and reproducibility

All images shown are representative of at least ten examples. No statistical method was used to predetermine sample size but we used prior experience to establish minimal sample sizes. No samples were excluded from analysis, provided staining was of high quality (exclusion of live imaging FUCCI samples with no cell-cycle phase transitions is described in Results).

Investigators were not blinded during outcome assessment, but had no pre-conception of what the outcomes might be. For MARCM studies, the EdU index was calculated for marked FSCs in each layer, for r1 and r2a ECs. For FSCs, we then normalized these values using controls because EdU incorporation among controls shows some variation between experiments. We calculated average EdU incorporation indices for each layer over all controls (35 in total) from many MARCM experiments. Layer 1 values are the most reliable because they derive from the most cells and the absolute EdU index is highest. We therefore multiplied the EdU index for each layer of an experimental genotype by the ratio of the layer 1 index value for all controls divided by that value for the control in the specific experiment to derive a normalized EdU index for each layer (these operations can be seen in the Fig 6 RawFSCdata tab of S1 Data). For genotypes tested in more than one experiment, the normalization was performed before taking the average of all normalized EdU indices in each layer. There was no normalization for EC EdU indices, which are zero for most genotypes, including controls. The "N-1" Chi-squared test method was used to calculate a Z score for determining significance of any differences between indicated genotypes for cells in the same location, and error was reported as standard error of a proportion. To determine whether the EdU index distribution among the FSC layers of an altered genotype differed significantly from controls, we first calculated the average EdU index for all FSCs of the altered genotype, with each layer contribution weighted based on the normal distribution of FSC among layers measured in appropriate controls. This average EdU index was then multiplied by the control EdU index for each layer to derive expected EdU indexes for each layer of an altered genotype if the EdU pattern matched controls. Finally, a chi-squared test was applied to compare observed and expected EdU indexes for each layer to determine the statistical significance of differences. Graphs of EdU indices also include tabulation of the average number of marked FSCs at 12d, the percentage of marked FSCs present in layer 1, and the probability (p) of a marked layer 1 FSC becoming an FC in a single 12h egg chamber budding cycle, calculated as previously documented [18]. From the latter two parameters, we calculated the percentage change in conversion of FSCs to FCs relative to wild-type per budding cycle, as described below.

The loss of a marked FSC of a specific genotype by differentiation to an FC each cycle,
F = f (fraction of FSCs in layer 1) × p(differentiation to FC) per marked FSC.
For WT FSCs, F = 0.48 × 0.64 = 0.307
The change in FSC conversion to FCs per FSC per cycle for a different FSC genotype,
$\Delta F = (fp—0.307)$.

## Supporting information

**S1 Methods. The diagrams and text in "Live FUCCI Calculation Method" and "Sampling considerations for Live FUCCI Calculation Method" provide an intuitive and mathematical explanation of how sampling of multiple cells during live imaging with FUCCI reporters provides data suitable for calculating the absolute duration of G1 and G2 phases.** (PDF)

**S1 Fig. *C587-GAL4* expression pattern and scoring cell cycle phases using FUCCI reporters and DAPI staining.** (A) Maximum projection of z sections spanning 15 μm of a C587>UAS-RFP germarium to show the *C587-GAL4* expression pattern (red) relative to Fas3 staining (white). (B) Single z-section of the same germarium with the anterior border of strong Fas3 staining indicated with a blue dotted line, allowing designation of FSCs in layers (labeled as 1, 2 or 3). ECs and FSCs express RFP strongly, with weaker expression in Layer 1 FSCs and occasional weak expression in immediate FCs. (C-H) A *C587>FUCCI* germarium stained for DAPI (white) to show all nuclei. The anterior border of strong Fas3 (blue) expression is

outlined by a blue dotted line in (D-H). FSCs in this germarium are in G2 (yellow arrows; GFP and RFP) or S phase (red arrow; no GFP or RFP). (C) Maximum projection of three z slices (z6-8) taken 2.5 μm apart, showing five FSCs. Three of these FSCs are shown in single-section images for z8 (D, E) and two others in z6 (F). Another FSC is visible only by examining z7 in isolation (G). (D) has only GFP and Fas3 channels, while (E) has only RFP and Fas3 channels, clarifying the presence of both GFP and RFP in the three indicated FSCs. (F-H) Individual, adjacent z sections showing germline cysts highlighted in different colors. Cysts are identified by clustered round germline cells which are larger than somatic cells. Identification of all germline nuclei allows even somatic nuclei with no GFP or RFP signal to be identified, including the layer 1 FSC in S-phase (red arrow) in (G). Most FSCs span two adjacent z sections: the Layer 2 cell in z6 (F) is also indicated in z7 (G). Note that layer 1 FSCs in (C-H) are immediately anterior to the anterior Fas3 border. Layer 2 FSCs are displaced roughly one cell body further anterior (left) and, in these examples, have a somatic cell nucleus between them and the Fas3 border. The single indicated layer 3 FSC is one cell diameter further anterior, neighboring a layer 2 FSC. Scale bar applies to all images; 20μm.
(PDF)

**S2 Fig. Similar patterns of initial cell labeling for H2B-RFP and GFP driven by the same promoter.** (A-C) Flies with *UAS-H2B-RFP*, *UAS-GFP*, *actin-GAL4* and *tub-tsGAL80* were kept at 29C for 4d and then fixed to show initial expression levels before any chase period at 18C. Both H2B-RFP (red) and GFP (green) showed strong expression in ECs and layer 2 and 3 FSCs (blue and yellow arrows), weaker expression in layer 1 FSCs (white arrows) and even weaker expression in FCs (pink bracket). Three consecutive middle z slices were combined for (A) RFP and GFP together, (B) RFP alone, and (C) GFP alone. Scale bar, 20μm.
(PDF)

**S1 Movie. FUCCI cell cycle reporter in a wild-type germarium.** Entire movie for germarium featured in Fig 4A for z-sections including the layer 1 FSC (arrow) that progresses from G2 (GFP + RFP) through mitosis to give two daughters in G1 (GFP), which progress to S-phase (no GFP) at different times. Images were collected every 10 min. Time is in hour:min.
(MOV)

**S2 Movie. FUCCI cell cycle reporter in a *C587-Hop* germarium with increased JAK-STAT pathway activity.** Entire movie for germarium featured in Fig 4B for z-sections including the layer 3 FSC (arrow) that progresses from G2 (GFP + RFP) through mitosis to give two daughters in G1 (GFP). Images were collected every 15 min. Time is in hour:min.
(MOV)

**S3 Movie. FSCs move in both directions between layers 1 and 2.** A germarium expressing *UAS-H2B-RFP* and *ubi-GFP* 21d after shift to 18C. Initial layer assignment of FSCs was determined by the relative strength of RFP expression. A layer 1 FSC expressing *ubi-GFP* and very low H2B-RFP (blue arrow) moved to layer 2. A layer 2 FSC expressing H2B-RFP and *ubi-GFP* (white arrow) moved to layer 1. Images were collected every 5 min. Time is in hours:min
(MOV)

**S4 Movie. A layer 2 FSC moves to EC region 2a.** Germarium expressing *UAS-H2BRFP* and *ubi-GFP* 16d after shift to 18C. Initial layer assignment of FSCs was determined by the relative strength of RFP expression. A layer 2 FSC (white arrow) moved to layer 3 at 45 min, and later moved anterior to region 2a. Images were collected every 9 min. Time is in hours:min.
(MOV)

**S5 Movie. A layer 3 FSC moves to layer 1.** Germarium expressing *UAS-H2BRFP* and *ubi-GFP* 16d after shift to 18C. Initial layer assignment of FSCs was determined by the relative strength of RFP expression. A layer 3 FSC (purple arrow), initially anterior to a layer 2 FSC (red arrow) and a layer 1 FSC (cyan arrow) moved past the layer 2 FSC to layer 1. Images were collected every 5 min 20 sec. Time is in min:sec.
(AVI)

**S1 Data. Each tab on this spreadsheet presents the raw data underlying the named Figure.**
(XLSX)

## Acknowledgments

We thank Burcu Gulez for contributions to H2B-RFP studies, Bruce Edgar (University of Washington, Seattle), Laura Buttitta (University of Michigan, Ann Arbor), and Bob Duronio (University of North Carolina, Chapel Hill) for reagents, Ray Sidney for discussion of mathematical modeling of the cell cycle, the Bloomington stock center for provision of genetic reagents, the Developmental Studies Hybridoma Bank (DSHB) for antibodies, FlyBase as an information resource, and the confocal microscope resource provided by the Department of Biological Sciences, Columbia University.

## Author Contributions

**Conceptualization:** David Melamed, Daniel Kalderon.

**Data curation:** David Melamed, Aaron Choi, Amy Reilein, Daniel Kalderon.

**Formal analysis:** David Melamed, Simon Tavaré, Daniel Kalderon.

**Funding acquisition:** Daniel Kalderon.

**Investigation:** David Melamed, Aaron Choi, Amy Reilein.

**Methodology:** David Melamed, Aaron Choi, Amy Reilein, Daniel Kalderon.

**Project administration:** Daniel Kalderon.

**Software:** Simon Tavaré.

**Supervision:** Amy Reilein, Daniel Kalderon.

**Validation:** David Melamed, Aaron Choi, Amy Reilein, Daniel Kalderon.

**Visualization:** David Melamed, Aaron Choi, Amy Reilein.

**Writing – original draft:** Daniel Kalderon.

**Writing – review & editing:** David Melamed, Amy Reilein, Simon Tavaré, Daniel Kalderon.

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
