## [Decision Letter · Decision Letter 0]

4 Jan 2023

Dear Dr Kalderon,

Thank you very much for submitting your Research Article entitled 'Spatial regulation of Drosophila ovarian Follicle Stem Cell division rates and cell cycle transitions' to PLOS Genetics.

The manuscript was fully evaluated at the editorial level and by independent peer reviewers. The reviewers appreciated the attention to an important problem, but raised some substantial concerns about the current manuscript. Based on the reviews, we will not be able to accept this version of the manuscript, but we would be willing to review a much-revised version. We cannot, of course, promise publication at that time.

If you decide to revise the manuscript for further consideration at PLOS Genetics, please aim to resubmit within the next 60 days, unless it will take extra time to address the concerns of the reviewers, in which case we would appreciate an expected resubmission date by email to plosgenetics@plos.org.

As you will see in the specific comments of the reviewers, there is a mixed view on the suitability of this manuscript for publication in PLOS Genetics. In addition to the detailed feedback provided by the reviewers directly to the authors, they raised several points in their comments to the editors that are important to address and which I summarize below:

1) All three reviewers found the manuscript an exceptionally difficult read, and they expressed concerns that this would lessen the impact of the work after publication. One reviewer suggested that the MARCM section of the manuscript could be deleted; another suggested limiting the MARCM data to the JAK/STAT pathway (because of the lack of information on the spatial levels of PI3K and Hippo signaling). This reviewer also suggested moving much of the H2B-RFP dilution data to the supplement. Besides concerns about the clarity/accessibility of the manuscript, two other concerns in the reviewers’ comments to the editor were as follows:

2) The reviewer who raised concerns about the spreadsheets for Figure 6 was motivated to look at the raw data to learn more about possible variability in the percent positive Edu cells between each germarium. Providing more information in this regard would be helpful.

3) One reviewer was surprised there was no mention in the manuscript concerning the controversy between your published work and that of Todd Nystul. Would be a service to readers to discuss this briefly in the introduction.

We are sorry that we cannot be more positive about your manuscript at this stage. Please do not hesitate to contact us if you have any concerns or questions.

Yours sincerely,

Ken M. Cadigan

Academic Editor

PLOS Genetics

Gregory P. Copenhaver

Editor-in-Chief

PLOS Genetics

Reviewer's Responses to Questions

**Comments to the Authors:**

Reviewer #1: The study by Melamed et al investigates the important question of how stem cell proliferation rates are regulated, using the Drosophila follicle stem cells as a stem cell model. These stem cells exhibit population asymmetry and heterogeneous behaviors depending on their position in layers 1, 2 or 3. Although their differential proliferation rates have been previously documented by this group, how these different proliferation rates are governed and maintained is still unknown, and this paper addresses that question, using several methods including FUCCI imaging in fixed and live samples, dilution of a fluorescent protein, and clonal analysis. The paper was previously submitted to PLOS Biology, where it was rejected and sent with its prior reviews to PLOS Genetics.

Unfortunately, the study has many serious methodological issues. Whether they can be fixed is not clear. Even without the problems in experimental design and missing data, the manuscript was so difficult to read that its greater significance would be lost on the reader. Thus, although my main concerns (see below) are more significant than writing style, clarity is also a major issue that needs addressing.

The first problem is that it seems to be difficult to recognize FSCs with precision. For example, in Fig. S1, the central green FSC, labeled as row 3, sits above and below cells that look identical to it and are at the same distance from the Fas3 border -- yet are not labeled as FSCs. How are such identifications made? There are very few examples of the actual image data that underlie this paper, and they give pause to whether the goals of this study can be met if FSCs are not identified by clonal methods. The authors must convince the readers that this is possible to do, especially important given the recent disagreements about the number and locations of the FSCs. In the methods they say that FSCs are identified by location, but that does not explain the cells above.

If the question of identifying FSCs can be addressed, the next question is whether it is possible to accurately assign FSCs to layers 1,2,3. In Fig. S1, the uppermost three FSCs are assigned to layers 1 (top), 1 (middle), 2 (bottom). Yet the top and bottom FSCs seem to have equal claim to being in layers 1 and 2, although assigned differently, because each is partially blocked by the middle FSC and partially open to the Fas3 border. If there is no unambiguous way to assign them to layers, the analysis could still have value, as the somewhat arbitrary assignment still will reflect a trend (as a layer 1 FSC is not likely to be assigned to layer 3, and vice versa), but it will not have the kind of precision the authors use throughout the paper.

The next issues concern data reporting, and these are serious issues. A previous reviewer asked the authors to “include n number in figure legends.” The authors responded, “We consider it impractical to cite those numbers in the legends or on the bars of graphs because of space constraints. We do supply spreadsheets with the raw numbers, together with the percentages displayed in bar graphs, as well as error and significance calculations, which of course utilize the n values.” This response seems disingenuous, considering the length and wordiness of the manuscript; n values take only a few lines to report at most. But worse, the data are not reported in the supplement. I discovered this when trying to understand how the cell cycle time is calculated from the data. The authors state, “if we add together the total time spent by all cells of a specific type… and divided by the number of transitions observed out of that phase we could infer the average phase duration”. This is not at all clear to me. To try to understand, I looked for the actual data in the supplement, but it is not present. (Tangentially, the author’s summary of the weaknesses of the previous approach was also not clear: “The fraction of cells in M-phase or S-phase under different conditions will, however, only provide a measure of the relative cell cycle times if the length of that phase remains constant.” Please clarify, perhaps illustrating this statement with an example of how it could lead to errors.)

The next data issue is the new technique of H2B-RFP dilution. It is a new assay using new reagents, yet it is not described or validated sufficiently for the reader to have confidence in it. The all-off (18°) and all-on (29°) conditions need to be shown in a supplement. The authors note that “the pattern of labeling intensities among different cells were very similar to those of GFP from a UAS-GFP transgene present in the same animals” but that data is missing. There are a great many qualitative statements without quantitative data to back them up. The most problematic is this: “Under all tested conditions, we observed uneven H2B-RFP (and GFP) intensities among germarial cells”. Without providing evidence about the distribution of these uneven intensities immediately after labeling, we cannot evaluate the significance of any change observed later that is attributed to dilution – we need the variability of both the starting conditions and the later dilution conditions, and we need to see if any difference in these highly variable numbers are statistically significant. This analysis will require not just chi-squared tests, comparing observed to expected, but statistical tests that will consider data variability. Other vague statements: “H2B-RFP signal was largely cleared…” [please show and quantify], “FSCs retained a strong H2B-RFP signal, while intensity in some layer 1 FSCs was markedly lower but clearly detectable” [how many layer 1 FSCs?].

Smaller points:

• In Fig. 1A, indicate R1 and R2A. Also R2a and R1 are capitalized in Fig. 3 but never in the text; standardize them.

• Show the expression pattern of C564Gal4 in a supplementary figure. Its gradient is discussed in the text several times, and it is a key reagent.

• In Fig. 6A, the diagram is confusing, and the legend does not help sufficiently. Same for the numbers above the chart in panel D – please label the 4 different numbers on the figure because the legend is very confusing (same with the similar charts in Figs. 7,8,9)

Reviewer #2: In this MS, the AUs report a thorough analysis of the cell cycle of live FSCs utilizing the FlyFUCCI tool and other genetic manipulations. They address the proliferation differences found in the 3 FSC layers and investigate the nature of the signals that may generate the diverse proliferative behaviors found in the FSC population. The AUs have made a significant effort to assemble a more tractable and easier to read MS (compared to the original submission). Still, it is considerably dense and detailed in its descriptions and analyses. In my opinion, the data presented are rigorous and comprehensive, and accompanied by pictures of great quality.

1- My main concern is the adscription of colorless (non-GFP, non-RFP) cells to the S-phase. In the text, the AUs state that “cells with GFP-only or GFP-plus-RFP expression did not include EdU, consistent with assignment of all such cells to G1 and G2 (or M), respectively (Fig. 1C-H). Some EdU-positive cells included RFP but the majority were colorless, showing that the delay in accumulating RFP in S-phase is extensive. All RFP-only cells had EdU label, indicating that GFP accumulates quickly at the onset of G2. Thus, G1 (GFP only), G2/M (GFP + RFP) are clearly defined, while S-phase cells can be recognized as lacking both GFP and RFP”. There is however at least one precedent in which other AUs conclude that non-RFP non-GFP FLyFUCCI cells are either in G1 or S phase (Villa-Fombuena et al, Dev. 2021). So my question for the AUs is whether there are colorless cells that are EdU negative? And in what frequency? This is an important point, particularly so considering the number of observations in which they quantify cell cycle phases considering these cells as S-phase. In addition, the very short G1 phase compared to the long S-phase described for most of the cell types and experimental situations (see for instance Fig. 4C) is striking when compared to other stem cell types.

2- Another general comment. I believe it would help assess the significance of the findings if the AUs were to indicate the sample size in the different quantifications. Perhaps not for all of the genotypes or experimental conditions, but to cite a range (for instance, n>20 for all of the genotypes) and perhaps not in the text or in the fig. itself, but in the fig. legends?

3- The text citation of Figs. 2, 3 and 4 is rather confusing. The citation order in the text is the following: figs. 2 + 3 (line 177), 3A-C (191), 4 (194), 2A + 3 (205), 2B + 3 (215), 3D (225), 2D + 3A-C (234), 2E + 3A-C (256)… I have struggled a bit to follow the text and the corresponding figs. This could be improved, perhaps by reorganizing Figs 2 and 3 and moving the EC data to a new Suppl fig.?

4- Line 101: Please introduce the concept of FSC depletion. Is there a Ref. for the 4-fold higher depletion rate of layer 1 FSCs?

5- Line 401: See my comment above RE colorless and S-phase.

6- Lines 662-664: “Live imaging with FUCCI reporters allowed us to record the frequency of cell-cycle transitions and deduce absolute cell cycle times. These parameters have not previously been measured for FSCs and are generally not known in other adult stem cell paradigms.” You may want to consider citing here Villa-Fombuena et al. These AUs precisely utilize FlyFUCCI to look at female Drosophila germline stem cells in live ovaries.

Reviewer #3: In this revised paper the authors examine the cell cycle regulation of FSC layers, which give rise to both Escort Cells and Follicle Cells, which have distinct cell cycle parameters, numbers and positions. They use FUCCI imaging (both live and fixed analysis), EdU labeling (with 1 hour labeling intervals) Histone-RFP depletion assays (to assess proliferative heterogeneity in FSCs) and MARCM clonal analysis to examine how cell cycle and signaling pathway manipulations impact the cell cycling parameters of FSCs. In brief they uncover: heterogeneity in the FSC cycling population such that layer 1 (giving rise for FCs) cycles fastest with a significant fraction in G2, layer 2 more slowly and layer 3 most slowly. They establish signals (specifically Jak/Stat and PI3K, with lesser data on Hippo signaling) that impact these transitions and identify CyclinE and Stg as the most likely downstream cell cycle machinery modulated by these signaling pathways.

One limitation of this story is that much of the evidence for the epistasis is quite indirect and thus the pathways are left somewhat unclear. For example, Dacapo over expression partially limits the cell cycle effects of Hop OE, but it is not clear whether Hop OE increases CycE expression, impacts Stg expression, or how exactly it affects G1 and (mostly) G2 timing. Similarly, the clonal analysis via MARCM shows partial rescue of Stat loss and cell cycle arrests by overexpression of CycE and Stg, but clearly there are unaddressed feedback loops that make the epistasis, even with CycE and Stg overexpression, inconclusive (e.g. adding CycE and Stg cannot fully restore cycling in the absence of Stat, which the authors point out). In addition, the regulation upstream or downstream of PI3Kinase activity and the Hippo pathway is unknown/unaddressed beyond the most downstream cell cycle regulators CycE and Stg. Altogether this provides detailed data on how specific genetic manipulations alter the cell cycle in FSCs, but does not focus on a specific pathway, and leaves unclear what exactly PI3K, Jak-Stat and Hippo pathways are doing to modulate FSC cell cycling. That said, I recognize there are limits to any one study and there has already been significant revision to shorten the paper. I therefore will try to limit my comments for further revision to suggestions to streamline the story or clarify issues.

One strength is that the authors carefully address the limitations of each approach to measuring the cell cycle parameters, which I appreciate. For example, fixed imaging approaches fail to capture dynamics and fail to accurately measure lengths of cell cycle phases. However a fundamental flaw in the live imaging of FUCCI cells is the loss of color during S-phase leading to an inherent inability to track cells throughout the cell cycle. Thus both approaches must be used together to confirm the cell cycle parameters. A limitation of the H2B-RFP dilution imaging was an inability to equalize starting amounts of H2B-RFP expression prior to tracing. The authors properly address each of these limitations (which is appreciated since these are often ignored), and use a combination of estimates from each cell cycle transition and its frequency to arrive at estimates of cell cycle phase timing for layer1 and layer 2 FSCs that roughly recapitulate estimates from fixed imaging. (They are unable to measure dynamics for layer 3 FSCs.) They uncover interesting differences in S-phase length between layers of FSCs, (which might suggest differences in E2F activity and therefore differences in levels of DNA synthesis/repair targets). This could possibly be addressed by looking at reporters of E2F transcriptional activity. In addition, I wonder if the regulation of PI3K activity could be addressed by looking at the tGPH reporter.

The H2B-RFP imaging description is complicated, due to the caveats that the authors point out. In sum, it supports the general finding of cell cycle speed from the live and fixed FuCCI experiments and I suggest it could largely be put into the supplement (and only briefly summarized as additional evidence supporting the FUCCI study) to address the length and complexity of the current (revised) manuscript. The authors describe the importance of this section being due to the evidence of FSC movement between layers - but I would suggest this portion showing FSC movement can be retained in the main manuscript, but without the additional details of the H2B-RFP dilution dynamics to simplify.

Clarification points:

Excess Cyclin E prolonging of S-phase can cause DNA damage and replication stress (see https://www.frontiersin.org/articles/10.3389/fcell.2021.774845/full and references therein). Is DNA damage incurred in FSCs in the cell cycle manipulations and thus reducing the subsequent proliferative capacity of FSCs when S-phase is prolonged? As mentioned above, if the signaling environment impacts the levels of E2F activity in each FSC layer, this could alter how the cells respond to CycE and Stg manipulations.

Page 86 line 199 - the minimal effects of CycE OE in anterior domains could be due to limiting Cdk2 levels (as opposed to different CycE/Cdk2 thresholds as the authors propose). The authors could examine this by co-overexpressing CycE and Cdk2 or could explain both possibilities.

The experiments blocking the Hippo signaling pathway in Stat mutant clones, with and without CycE/Stg expression seem to lack some context to explain how it fits into the revised story. If I understand correctly, the sum total of this experiment is that CycE & Stg are downstream of both pathways (already known) but that Hippo signaling and Stat loss still limit the level of FSC cycling that can be induced by CycE/Stg (e.g. effects of CycE/Stg expression are stronger when Stat is present but are partially enhanced when Kibra is also lost). Overall this suggests a non-linear pathway with additional targets affected by both Hippo and Stat signaling outside of CycE and Stg that feed into the response to the ectopic CycE and Stg. Should this and potential additional cell cycle regulatory targets of these pathways be discussed?

This is outside the scope of this paper, but I’m very intrigued by the stimulation of S-Phase entry seen by excess Stg in ECs. This suggests some cell type-specific cdc25a/b function for Stg which is interesting. I wonder if these cells are entering an endocycle rather than a mitotic cycle?

**Have all data underlying the figures and results presented in the manuscript been provided?**

Reviewer #1: **No: **see review and confidential comments to the editor

Reviewer #2: Yes

Reviewer #3: Yes

PLOS authors have the option to publish the peer review history of their article (what does this mean?). If published, this will include your full peer review and any attached files.

Reviewer #1: No

Reviewer #2: No

Reviewer #3: No

---

## [Decision Letter · Decision Letter 1]

13 Jun 2023

Dear Dr Kalderon,

Thank you very much for submitting your Research Article entitled 'Spatial regulation of Drosophila ovarian Follicle Stem Cell division rates and cell cycle transitions' to PLOS Genetics.

The manuscript was fully evaluated at the editorial level and by independent peer reviewers. The reviewers appreciated the attention to an important problem, but raised some substantial concerns about the current manuscript. Based on the reviews, we will not be able to accept this version of the manuscript, but we would be willing to review a much-revised version. We cannot, of course, promise publication at that time.

Should you decide to revise the manuscript for further consideration here, your revisions should address the specific points made by reviewer 4.  If you decide to resubmit, I am asking you to carefully consider reviewer 4’s thoughtful comments and address them in a constructive manner. We will also require a detailed list of your responses to the review comments and a description of the changes you have made in the manuscript.

If you decide to revise the manuscript for further consideration at PLOS Genetics, please aim to resubmit within the next 60 days, unless it will take extra time to address the concerns of the reviewers, in which case we would appreciate an expected resubmission date by email to plosgenetics@plos.org.

We are sorry that we cannot be more positive about your manuscript at this stage. Please do not hesitate to contact us if you have any concerns or questions.

Yours sincerely,

Ken M. Cadigan

Academic Editor

PLOS Genetics

Gregory P. Copenhaver

Editor-in-Chief

PLOS Genetics

Reviewer's Responses to Questions

**Comments to the Authors:**

Reviewer #2: All my comments have been addressed in depth. While the AUs have decided to follow some of my suggestions, but not others, the end result is a solid and thorough paper. The AUs should take the responsibility and decide for themselves the final organisation of the MS. Lets hope that their final arrangement of the data works out and the audience enjoy the read and appreciate the sound science in it.

Reviewer #4: See attachment

**Have all data underlying the figures and results presented in the manuscript been provided?**

Reviewer #2: Yes

Reviewer #4: Yes

PLOS authors have the option to publish the peer review history of their article (what does this mean?). If published, this will include your full peer review and any attached files.

Reviewer #2: No

Reviewer #4: No

---

## [Editor Report · Decision Letter 2]

11 Sep 2023

Dear Dr Kalderon,

We are pleased to inform you that your manuscript entitled "Spatial regulation of Drosophila ovarian Follicle Stem Cell division rates and cell cycle transitions" has been editorially accepted for publication in PLOS Genetics. Congratulations!

Yours sincerely,

Ken M. Cadigan

Academic Editor

PLOS Genetics

Gregory P. Copenhaver

Editor-in-Chief

PLOS Genetics

Comments from the reviewers (if applicable):

**Data Deposition**

http://datadryad.org/submit?journalID=pgenetics&manu=PGENETICS-D-22-01105R2

**Press Queries**

---

## [Editor Report · Acceptance letter]

20 Sep 2023

PGENETICS-D-22-01105R2 

Spatial regulation of Drosophila ovarian Follicle Stem Cell division rates and cell cycle transitions 

Dear Dr Kalderon, 

We are pleased to inform you that your manuscript entitled "Spatial regulation of Drosophila ovarian Follicle Stem Cell division rates and cell cycle transitions" has been formally accepted for publication in PLOS Genetics! Your manuscript is now with our production department and you will be notified of the publication date in due course.

With kind regards,

Judit Kozma

PLOS Genetics

On behalf of:
